# A high-energy sulfur cathode in carbonate electrolyte by eliminating polysulfides via solid-phase lithium-sulfur transformation

Xia Li[1], Mohammad Banis [1,2], Andrew Lushington[1], Xiaofei Yang[1,3], Qian Sun[1], Yang Zhao[1], Changqi Liu[1,3], Qizheng Li[1], Biqiong Wang[1,4], Wei Xiao[1,4], Changhong Wang[1], Minsi Li[1,4], Jianwen Liang[1], Ruying Li[1], Yongfeng Hu[2], Lyudmila Goncharova[5], Huamin Zhang[3], Tsun-Kong Sham[4] & Xueliang Sun[1]

Carbonate-based electrolytes demonstrate safe and stable electrochemical performance in lithium-sulfur batteries. However, only a few types of sulfur cathodes with low loadings can be employed and the underlying electrochemical mechanism of lithium-sulfur batteries with carbonate-based electrolytes is not well understood. Here, we employ in operando X-ray absorption near edge spectroscopy to shed light on a solid-phase lithium-sulfur reaction mechanism in carbonate electrolyte systems in which sulfur directly transfers to $Li_2S$ without the formation of linear polysulfides. Based on this, we demonstrate the cyclability of conventional cyclo-$S_8$ based sulfur cathodes in carbonate-based electrolyte across a wide temperature range, from $-20\,°C$ to $55\,°C$. Remarkably, the developed sulfur cathode architecture has high sulfur content (>65 wt%) with an areal loading of $4.0\,mg\,cm^{-2}$. This research demonstrates promising performance of lithium-sulfur pouch cells in a carbonate-based electrolyte, indicating potential application in the future.

[1] Department of Mechanical and Materials Engineering, University of Western Ontario, London, ON N6A 5B9, Canada. [2] Canadian Light Source, 44 Innovation Boulevard, Saskatoon, SK S7N 2V3, Canada. [3] Division of Energy Storage, Dalian Institute of Chemical Physics, Chinese Academy of Sciences, 116023 Dalian, China. [4] Department of Chemistry, University of Western Ontario, London, ON N6A 5B9, Canada. [5] Department of Physics and Astronomy, University of Western Ontario, London, ON N6A 3K7, Canada. These authors contributed equally: Xia Li, Mohammad Banis, Andrew Lushington. Correspondence and requests for materials should be addressed to X.S. (email: xsun9@uwo.ca)

Lithium–sulfur (Li–S) batteries are attractive candidates for the use in electric vehicles due to the ultra-high theoretical energy density[1,2]. However, state-of-the-art Li–S batteries utilize ether-based electrolytes that may face a series of challenges[3,4]. First of all, polysulfides, as the intermediate discharge products of Li–S batteries in ether-based electrolyte, are highly soluble in ether-based solvents and can easily transport from the cathode to the anode[3,4]. This phenomenon, referred to as the "shuttle effect", results in loss of active sulfur and corrosion of Li metal[5,6]. Moreover, ether-based solvents are highly volatile and have low flash points, thereby limiting battery application and posing a significant risk for batteries operating at elevated temperatures[7–10]. Therefore, despite the popularity of ether-based Li–S batteries, the practical use of this electrolyte system undeniably faces severe safety concerns.

Many of the issues described above can be circumvented by using a carbonate-based electrolyte. Carbonate-based electrolyte systems have been used in commercial Li-ion batteries (LIBs) due to their safe and stable properties as well as wide operation temperature window for nearly 30 years[11,12]. Furthermore, many flame-retardant additives designed for carbonate-based electrolytes have been investigated and applied into the battery market to further enhance their reliability[13,14]. Therefore, it is expected that a smooth transformation from the traditional metal-oxide cathodes of state-of-the-art LIBs to sulfur cathodes may promote the realization of safe and high-energy Li–S batteries with the mutual carbonate electrolyte system in the future. Actually, previous reports of carbonate Li–S batteries have demonstrated enhanced safety and stable cycling performance[15–17]. However, almost all of the carbonate-electrolyte-based Li–S batteries in previous references require unique sulfur cathodes with delicate synthetic procedures in order to achieve reversible Li–S reactions. A few commonalities exist across the literature for the sulfur cathodes in carbonate-based electrolyte: (1) confinement of short-chain sulfur molecules within a microporous structure or strong chemical bonding to a polymeric host, resulting in (2) very limited sulfur mass content (mostly <40 wt% in the whole electrode) and minimal areal loading[17–19]. According to these characteristics, there is widespread consensus among researchers that the success of carbonate Li–S batteries relies on the short-chain sulfur cathodes that is inherently accompanied with low sulfur loading. As a result, the low sulfur loading and complicated architecture design of these carbonate-viable sulfur cathodes severely diminish their application. There are very few reports that demonstrate a reversible Li–S electrochemical process in carbonate electrolyte with high sulfur loading cathodes in a conventional cyclo-$S_8$ molecule format.

In this study, synchrotron-based in operando X-ray absorption near-edge spectroscopy (XANES) is conducted to elucidate detailed mechanisms of Li–S batteries operated in both ether- and carbonate-based electrolyte. Compared to conventional ether-based electrolyte, in operando XANES reveals a drastically different electrochemical reaction pathway for Li–S cells in carbonate-based electrolyte. Interestingly, evidence of formation of linear polysulfides was absent in the spectra, suggesting a direct solid-phase transition of sulfur (both cyclo-$S_8$ and short-chain sulfur) to $Li_2S$. This fundamental mechanistic study indicates that the success of Li–S batteries in carbonate-based electrolyte is not determined by the allotrope of sulfur but rather by the electrochemical reaction pathway undertaken. Based on the reaction mechanisms elucidated in this study, conventional carbon–sulfur (C–S) electrodes with cyclo-$S_8$ molecule are developed and the electrodes display excellent electrochemical performance in a wide temperature range from −22 to 55 °C. Furthermore, sulfur cathodes with high sulfur content (67 wt% in sulfur composites) and high areal loading (4.0 mg cm$^{-2}$) exhibit stable capacity over

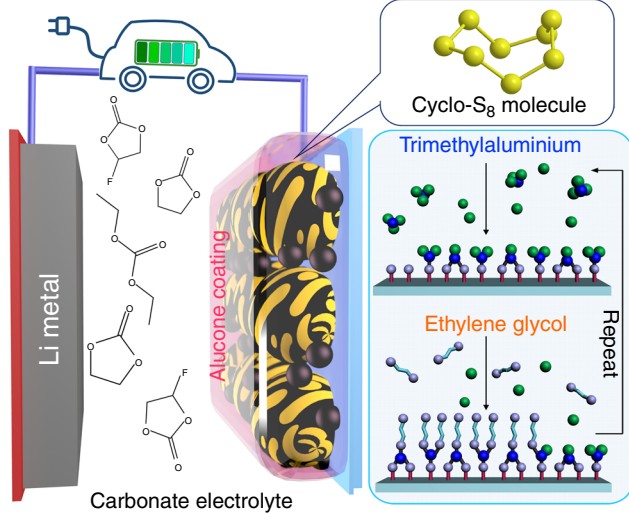

**Fig. 1** Schematic of a lithium sulfur battery in carbonate-based electrolyte. Alucone coating is applied to carbon–sulfur electrodes and the sulfur cathode is in cyclo-$S_8$ molecule format. Alucone thin film is directly deposited on the C–S electrodes by alternatively introducing trimethylaluminium and ethylene glycol via molecular layer deposition. Blue balls represent aluminium, green ball represent methyl, and gray balls represent hydroxyl

300 cycles. In particular, we conduct the pouch cell test of Li–S battery in carbonate-based electrolyte and measure the energy density. The development of high loading sulfur cathodes and Li–S pouch cells are a revolutionary breakthrough to the traditional low content sulfur cathodes in carbonate-based electrolyte, paving a new future for the development of safe and high-energy Li–S batteries.

## Results

**In operando XANES study of Li–S reaction mechanism**. Figure 1 provides a schematic outline for the configuration of a Li–S cell operating in carbonate-based electrolyte (carbonate Li–S cell) using an alucone-coated C–S cathode. In our previous study, we demonstrated that an alucone-coated commercial C–S cathode can reversibly cycle in carbonate electrolyte[7]. Alucone films are deposited using molecular layer deposition (MLD). This technique employs the use of self-limiting gas-phase reactions to produce ultrathin and conformal films[20–24]. Supplementary Figs. 1, 2 present field emission scanning electron microscopic (FE-SEM) images of commercial carbon (KJ-EC600)–sulfur electrodes without and with 10 cycles alucone coating along with elemental mappings of the electrode. The morphology of the coated C–S electrode is nearly identical to the pristine one, with particle sizes in the range of 40–60 nm. Supplementary Figs. 2b, c demonstrate the uniform distribution of MLD alucone, further illustrating the conformal growth process of this deposition technique[7].

The successful cyclability of C–S electrodes in carbonate electrolyte inspired several questions regarding the underlying mechanism governing this process: (1) as shown in Fig. 2a, b, the alucone-coated C–S electrodes cycled in ether-based electrolyte display a drastically different discharge–charge profile compared to the one in carbonate-based electrolyte. In ether-based electrolyte, a similar discharge–charge profile reported in the literature is observed. This profile typically displays two discharge plateaus and is thought to stem from the reduction of cyclo-$S_8$ molecule to longer-chain polysulfides (2.3 V) and then to short-chain sulfides (2.1 V)[25,26]. However, in carbonate-based electrolyte, the alucone C–S cathode presents a single discharge–charge

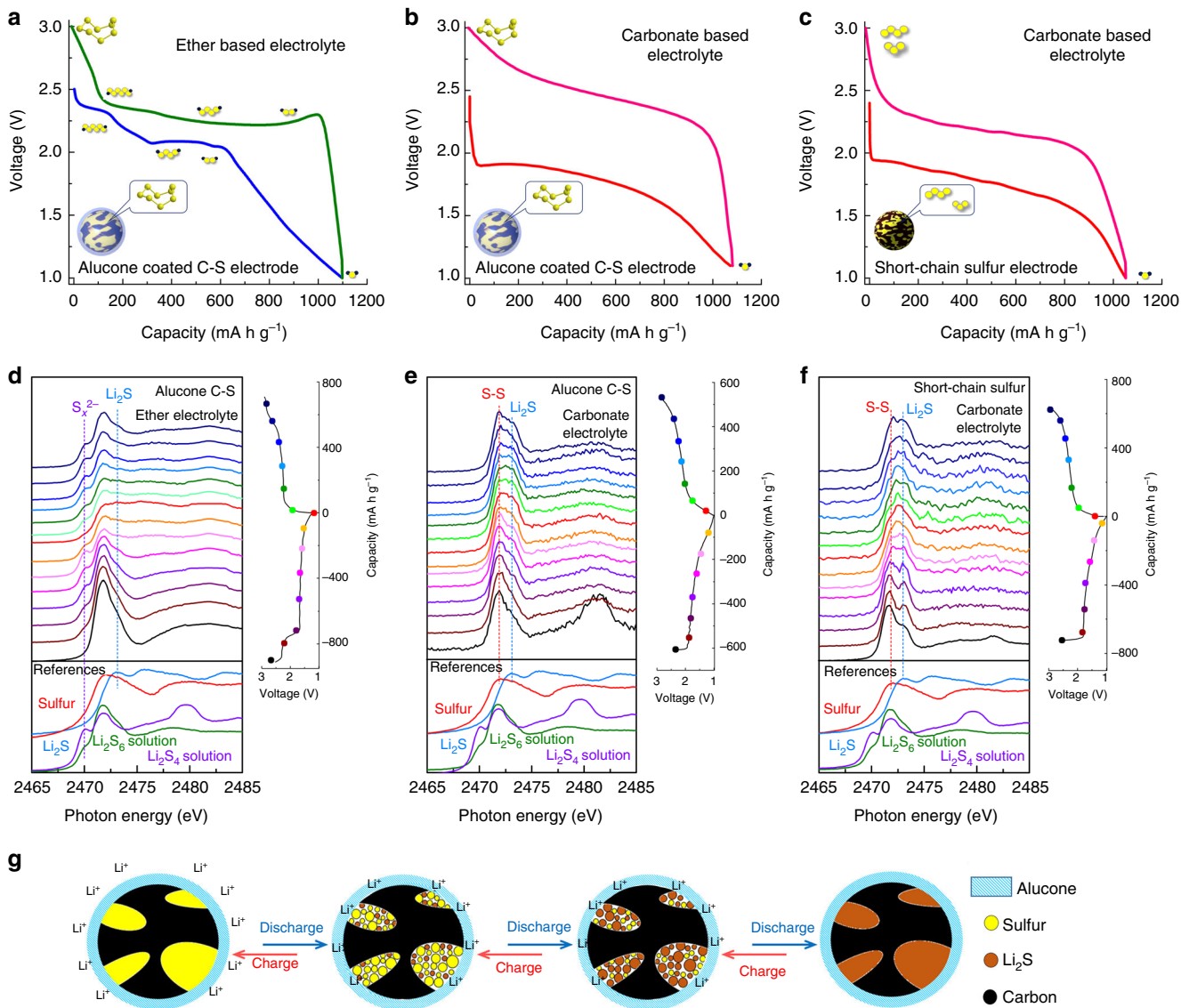

**Fig. 2** Understanding the reaction mechanisms of different lithium–sulfur cells. **a**–**c** Discharge–charge profiles of different types of lithium–sulfur cells. In operando X-ray absorption near-edge spectroscopy study of **d** alucone-coated C–S electrode in ether-based electrolyte, **e** alucone-coated C–S electrode in carbonate-based electrolyte, and **f** as-prepared short-chain sulfur electrode in carbonate-based electrolyte. **g** Schematics for proposed mechanism of alucone C–S cathodes in carbonate-based electrolyte

plateau. This then poses the question, do alucone-coated sulfur cathodes undergo an alternate Li–S electrochemical reaction in carbonate-based electrolyte than in ether-based one? (2) The reversible Li–S redox reaction of alucone-coated C–S electrodes, shown in Fig. 2b, demonstrates the possibility of conventional cyclo-$S_8$ cathodes operating in carbonate-based electrolyte. To the best of our knowledge, the majority of reported literature suggests that the reversible cyclability of Li–S cells in carbonate-based electrolyte is rooted in using cathode architectures that employ short-chain sulfur molecules, as shown in Supplementary Table 1 (small sulfur, polyacrylonitrile–sulfur composites, etc). Short-chain sulfur is in a metastable state and is formed via confinement of microporous carbon or chemical bonding with polymer skeleton[16,27]. On the other hand, although the molecular format of sulfur in our alucone-coated C–S cathode (cyclo-$S_8$) is different from the reported short-chain sulfur cathodes in the literature, a similar discharge–charge profile is observed (Fig. 2b vs Fig. 2c)[15,16,28,29]. Thereby, do cyclo-$S_8$-based cathodes and short-chain sulfur cathodes undergo a similar Li–S

electrochemical redox reaction in carbonate-based electrolyte? If the molecular format of sulfur is not a decisive factor for electrochemical reversibility, then what is?

To address these questions, in operando XANES measurements were conducted in carbonate- and ether-based electrolytes for alucone-coated C–S electrodes and compared to a cathode using short-chain sulfur that was also cycled in carbonate-based electrolyte. The results of the operando sulfur K-edge XANES with reference samples are presented in Fig. 2d–f. Detailed operating parameters are outlined in the Methods section and Supplementary Fig. 3[30]. Closely observing the whiteline region for all spectra, a feature at 2472.0 eV is present and can be attributed to the S 1s to S–S π* state transition of elemental sulfur[31,32]. As shown in Fig. 2e, an additional feature at 2473.0 eV appears and gradually becomes stronger with continued lithiation. This new feature can be attributed to the S 1s to Li₂S σ* transition[33,34]. This evolution is observed to be reversible during the charging process. Interestingly, an additional feature appears at 2481.5 eV and is identified as sulfate species. This chemical moiety forms following

battery assembly. However, shortly after resting in a helium-filled chamber, the strength of this feature diminishes and then stabilizes during electrochemical cycling[35]. The formation of $SO_4^{2-}$ is not fully understood and may be related to side reactions occurring between alucone, sulfur, and the electrolyte during electrochemical cycling. It should be noted that the peak labeled as $Li_2S$ in Fig. 2e is not fully reversed back to sulfur during the charging process. The first cycle coulombic efficiency of alucone-coated C–S cathode is around 90%, which indicates that a portion of the $Li_2S$ is irreversibly lost. Therefore, it is reasonable that at the end of the first charging process the $Li_2S$ peak is still observed in the XANES spectra. Figure 2d presents the collected XANES data for alucone-coated C–S cathode cycled in ether-based electrolyte. The feature at 2470.1 eV can be assigned to the S $1s$ to $\pi^*$ state transition associated with linear polysulfides[31,33,36]. The intensity of this peak is found to vary as the electrochemical reaction proceeds and corresponds to the redox reaction of polysulfides. At the end of the discharge process, the linear polysulfide peak disappears while the peak of $Li_2S$ becomes more prominent. The evolution of the ether-based Li–S cell, shown in Fig. 2d, is similar to the ones reported in the literature, demonstrating that the well-known Li–S redox mechanism accompanied the formation of polysulfide intermediate products[33,36–38]. On the contrary, the sulfur K-edge XANES of alucone C–S electrodes cycled in carbonate electrolyte (Fig. 2e) does not exhibit the distinct features associated with linear polysulfides during the discharge–charge process. This indicates that the alucone C–S cathode cycled in carbonate electrolyte involves direct electrochemical conversion of cyclo-$S_8$ to $Li_2S$ without the formation of any other intermediary products[33,36,39]. The operando sulfur K-edge XANES, in Fig. 2e, demonstrates an alternate electrochemical reaction pathway for alucone C–S cathodes in carbonate-based electrolyte and sheds light on the observed single plateau behavior in the discharge–charge profile.

To further elucidate the electrochemical pathways of sulfur cathodes in carbonate-based electrolyte, operando sulfur K-edge XANES study of Li–S cells using short-chain sulfur cathodes was also conducted. Details regarding the synthesis as well as physical and electrochemical characterizations of short-chain sulfur cathodes are outlined in the Methods section and Supplementary Figs. 4–5[15,40,41]. As shown in Fig. 2f, XANES of short-chain sulfur cathodes presents a similar electrochemical evolution to alucone-coated C–S cathodes in carbonate electrolyte. The absence of the peak at 2470.1 eV suggests that polysulfide formation does not transpire. The comparison of the two sulfur cathodes (alucone-coated C–S and short-chain S) confirms that both systems undergo a similar electrochemical reaction of sulfur being directly transferred to $Li_2S$ without the formation of polysulfides species in carbonate electrolyte. These operando XANES studies allow us to confidently address the questions posed earlier that molecular allotrope of sulfur does not necessarily govern reaction reversibility of Li–S cells in carbonate-based electrolyte. Rather, our results insinuate that by engineering the surface of sulfur cathode, a solid-phase Li–S electrochemical reaction route is possible[42–44]. Figure 2g outlines a schematic diagram of the proposed electrochemical reaction for alucone-coated C–S cathodes in carbonate electrolyte. According to the collected electrochemical and physical results, MLD alucone coating forces sulfur cathodes to undergo a solid-phase Li–S redox reaction in carbonate electrolyte. During this solid-phase electrochemical process, elemental sulfur is directly transferred to $Li_2S$ and bypasses the formation of unwanted intermediary products. To further demonstrate the elimination of polysulfides, time-of-flight secondary ion mass spectrometry (TOF-SIMS) of alucone-coated C–S electrodes and Rutherford backscattering spectrometry (RBS) of cycled Li metal anodes were

carried out and shown in Supplementary Fig. 6. Additionally, physical observation of electrolyte following electrochemical reaction is also shown in Supplementary Fig. 6. Detailed experimental process and results can be seen in Methods section and supplementary information. To understand the relationship between cyclo-$S_8$-based cathodes and carbonate-based electrolyte, we design a group of solution based experiments, as shown in Supplementary Fig. 7. The reversibility of cyclo-$S_8$ cathodes in carbonate-based electrolyte may be related to the stability of polysulfide species within the medium. The experiment concludes that lithium polysulfides are highly unstable within carbonate electrolyte systems, resulting in decomposition and partial conversion of polysulfides to elemental sulfur. The conformal alucone coating for C–S composites, on the other hand, effectively circumvents the side reactions to form a large amount of sulfur precipitates in the electrolyte and therefore forces the sulfur encapsulated in conductive carbon to undergo a solid-phase Li–S redox reaction. This is believed to be the underlying mechanism that allows for a reversible electrochemical reaction to take place in carbonate-based electrolyte. It should be noted that the reactions between sulfur species and carbonate or ether-based electrolyte are very complicated and not totally understood. This area of research should be further investigated with theoretical calculations in the future[45–47].

**Electrochemical characterization of carbonate Li–S batteries.** Based on the revealed reaction mechanism, it is clear that electrolyte composition and Li-ion diffusion will play a significant role in solid-phase Li–S reactions[6,48]. In an attempt to improve the performance of alucone-coated C–S electrodes, several different carbonate-based electrolyte compositions were investigated. Figure 3a presents various carbonate electrolyte systems with the addition of fluoroethylene carbonate (FEC) to improve Li–S cell cycle stability. The inspiration to incorporate FEC stems from its widespread use in stabilizing the surface of the metallic Li anode. FEC has been shown to enhance the solubility of lithium and promote rapid Li transport, resulting in an enhancement of the lithiation process occurring at the cathode[8,49–51]. Therefore, carbonate Li–S batteries with FEC is hypothesized to improve cell performance. As shown in Fig. 3b, the discharge–charge profiles of the cells with FEC demonstrate reduced polarization as well as a characteristic flat discharge potential plateau, illustrating improved electrochemical reaction kinetics during lithiation. Electrochemical performance of Li–S cells, presented in Fig. 3c, d, can be further improved by optimizing the ratio of FEC in the electrolyte. Impressively, by using 20 vol% of FEC, a highly stable Li–S cell can be made, retaining a capacity of 670 mA h g$^{-1}$ over 100 cycles with an average capacity loss <0.11% from the second cycle. Interestingly, further increasing the amount of FEC in the electrolyte to 30 vol% provides little improvement as the electrolyte is already saturated with 20 vol% of FEC. To further demonstrate the effect of FEC, environmental SEM images of Li metal anodes were taken following discharge–charge cycling. As shown in Supplementary Fig. 8, the surface of Li metal cycled in carbonate electrolyte, without FEC, is littered with lithium fragments (Supplementary Fig. 8b). Furthermore, cross-sectional views of the metallic anode in Supplementary Fig. 8d reveal the presence of a thick interlayer covering the surface of Li, with deposition protruding deep into the metallic anode. This may result in reduced Li/Li$^+$ transformation and poor ionic conductivity. However, as shown in Supplementary Figs. 8a, c, the morphology of Li metal cycled with FEC appears to be relatively pristine and uniform with few fragments appearing on the surface. Based on these observations, FEC is determined to play an important role in promoting fast Li transport in carbonate-based

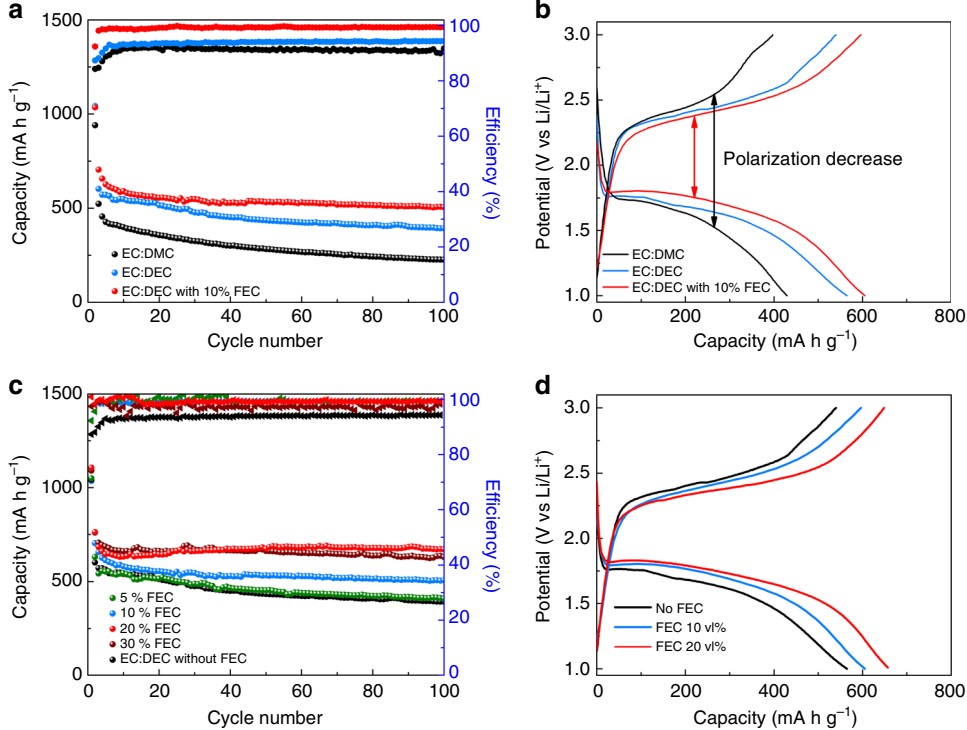

**Fig. 3** Optimization of electrolyte compositions for carbonate lithium–sulfur batteries. **a**, **b** Electrochemical performance of lithium–sulfur batteries with different carbonate electrolyte systems. The employed carbonate electrolyte systems are 1 M LiPF$_6$ in (black line) ethylene carbonate (EC) and dimethyl carbonate (DMC) with volume ratio of 1:1; (blue line) EC and diethyl carbonate (DEC) with volume ratio of 1:1; (red line) EC:DEC with 10 vol% fluoroethylene carbonate (FEC) additive. **c**, **d** Electrochemical performance of lithium–sulfur batteries with various ratios of FEC additive from 0 to 30 vol% in carbonate electrolyte (1 M LiPF$_6$, EC:DEC)

electrolyte and therefore can enhance the cycling stability and capacity of Li–S cells[6,25,52]. An optimized electrolyte composition with 20 vol% FEC is chosen as an ideal system for following electrochemical characterization.

In addition to the electrolyte composition, carbon host architecture plays an important role in Li–S batteries. In conventional ether-based Li–S batteries, due to the solid–liquid dual-phase Li–S redox reaction, a carbon host with large pore volume and fitted pore diameter is required to maintain an appropriate equilibrium between sulfur dissolution and retention[53–56]. For carbonate-based Li–S batteries, an appropriate carbon host for sulfur cathodes may require an alternate carbon architecture design. To demonstrate the universality and to further optimize the performance of sulfur cathodes with alucone coating, three commercial carbon materials (BP800, BP1300, and BP2000) with various porous structures are selected as hosts for sulfur cathodes. Supplementary Fig. 9 provides an outline for the surface analysis of these carbon materials. BP800 carbon is primarily composed of large mesopores in the range of 10–50 nm. On the other hand, the pore volume for BP2000 primarily originates from micropores and smaller mesopores. Detailed data of surface and pore properties of the three commercially available porous carbon materials are listed in Supplementary Table 2. These three carbon materials were then impregnated with sulfur, prepared into electrodes, and coated with 10 cycles of alucone. Figure 4a presents the cycle performance of these three alucone-coated C–S electrodes in the optimized carbonate electrolyte. Impressively, the three C–S electrodes demonstrate high electrochemical reversibility. This further exemplifies the universality of the alucone coating on porous C–S cathodes and allowing for electrochemical reversibility in carbonate electrolyte. Among the three C–S cathodes, BP2000 presents the most promising cycling

performance with an initial discharge capacity of 1160 mA h g$^{-1}$ and a capacity of 818 mA h g$^{-1}$ after 100 cycles. The differences in cycling performance between the three sulfur electrodes are proposed to originate from the carbon hosts. Compared to the other carbon hosts, BP2000 has the highest surface area with confined porous architecture. This unique structure allows highly dispersed sulfur distribution throughout the carbon host and improved electronic conductivity of sulfur in the solid-phase Li–S reaction (Supplementary Fig. 10). Electrochemical characterizations of bare C–S electrodes without alucone coating shown in Supplementary Figs. 11–12 illustrate the irreversible electrochemical process typically observed in carbonate electrolyte[57,58].

Various other electrochemical characterization techniques were also performed to further elucidate the performance of alucone-coated sulfur cathodes with the use of BP2000. Figure 4b displays rate performance of alucone-coated C–S electrodes. The electrode retains a capacity of over 550 mA h g$^{-1}$ at 1600 mA g$^{-1}$ and decreases to 320 mA h g$^{-1}$ when elevating the current density to 3200 mA g$^{-1}$. Figure 4c presents voltage curves for alucone-coated sulfur cathodes using galvanostatic intermittent titration technique (GITT). The curves present a pair of discharge–charge potential plateaus, indicating a single-phase electrochemical reaction. Cyclic voltammograms of the three alucone-coated C–S cathodes are presented in Fig. 4d and Supplementary Fig. 13. All coated cathodes demonstrate reversible Li–S redox electrochemical reaction in carbonate electrolyte, with a single pair of cathodic and anodic peaks at around 1.75 and 2.3 V, respectively. Compared to the other two carbon hosts, the sulfur cathode with BP2000 exhibits sharp and reversible peaks. Furthermore, the cathodic peak potential of the sulfur cathode with BP2000 is also higher than the other carbon architectures, indicating decreased polarization and elevated electrochemical activity. To evaluate the

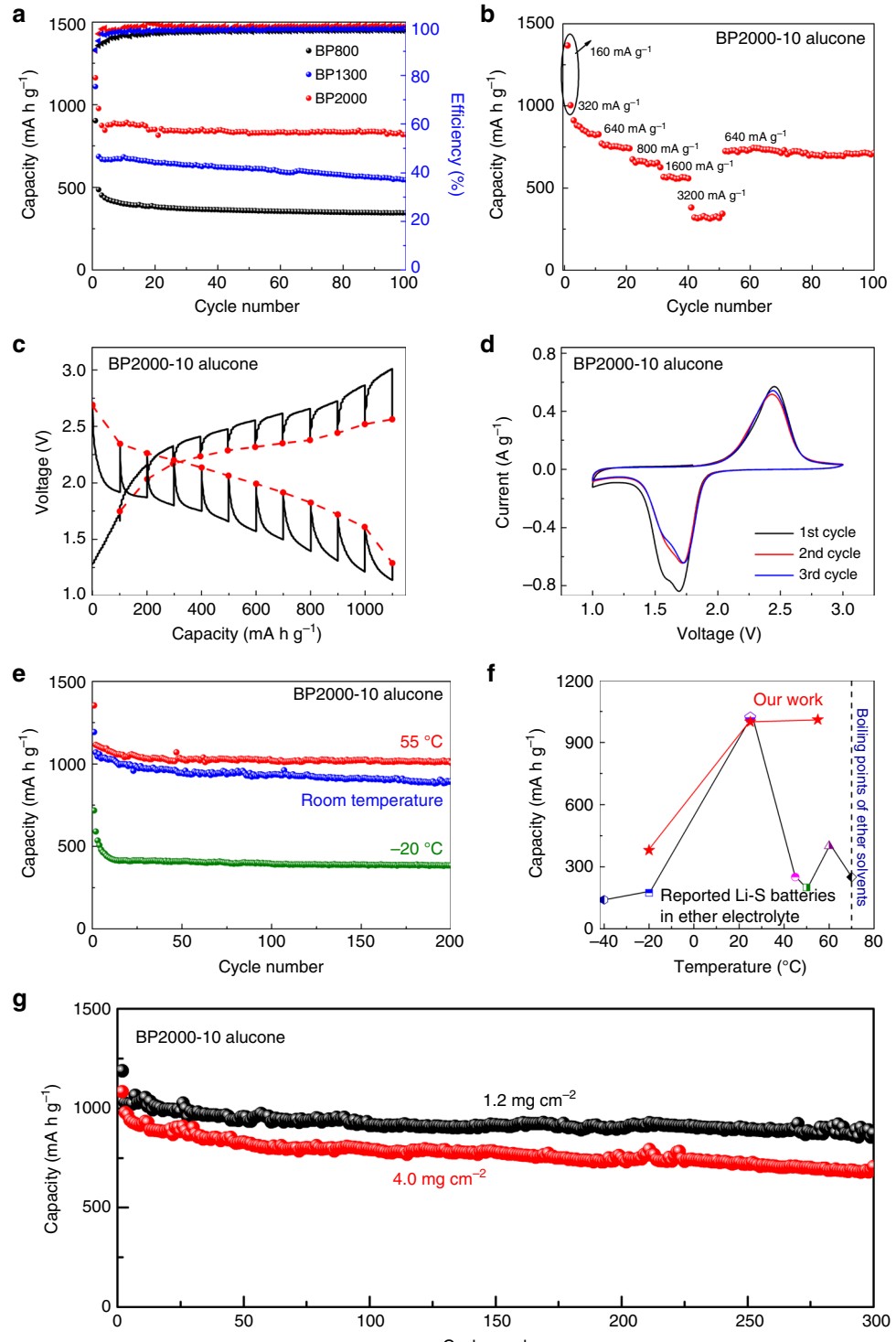

**Fig. 4** Optimization of carbon hosts for sulfur cathodes in carbonate lithium–sulfur cells. **a** Cycle performance of 10-cycle alucone-coated C–S electrodes with different carbon hosts at a current density of 320 mA g$^{-1}$. **b**–**g** Electrochemical characterizations of 10-cycle alucone-coated sulfur cathodes with BP2000 as carbon hosts (BP2000-10 alucone): **b** rate performance, **c** equilibrium voltage (red dashed lines) and transient voltage (black solid lines) profile vs. capacity, **d** cyclic voltammogram, **e** cycle performance operating at various temperatures, **f** comparison of reported Li–S cells and our work at various temperatures (Supplementary Table 3), and **g** long cycling performance of alucone-coated C–S electrodes with various sulfur loadings

practical application of carbonate Li–S batteries, cells were cycled at various temperatures, as shown in Fig. 4e. The batteries demonstrate stable cycling performance at elevated (55 °C) and low (−20 °C) temperatures and can retain a capacity of over 1010

and 380 mA h g$^{-1}$ at 160 mA g$^{-1}$ after 200 cycles, respectively. Compared to the performance of the reported Li–S cells (Fig. 4f and Supplementary Table 3), the developed carbonate-based Li–S cells demonstrate ultra-stable and prolonged cycle life at various

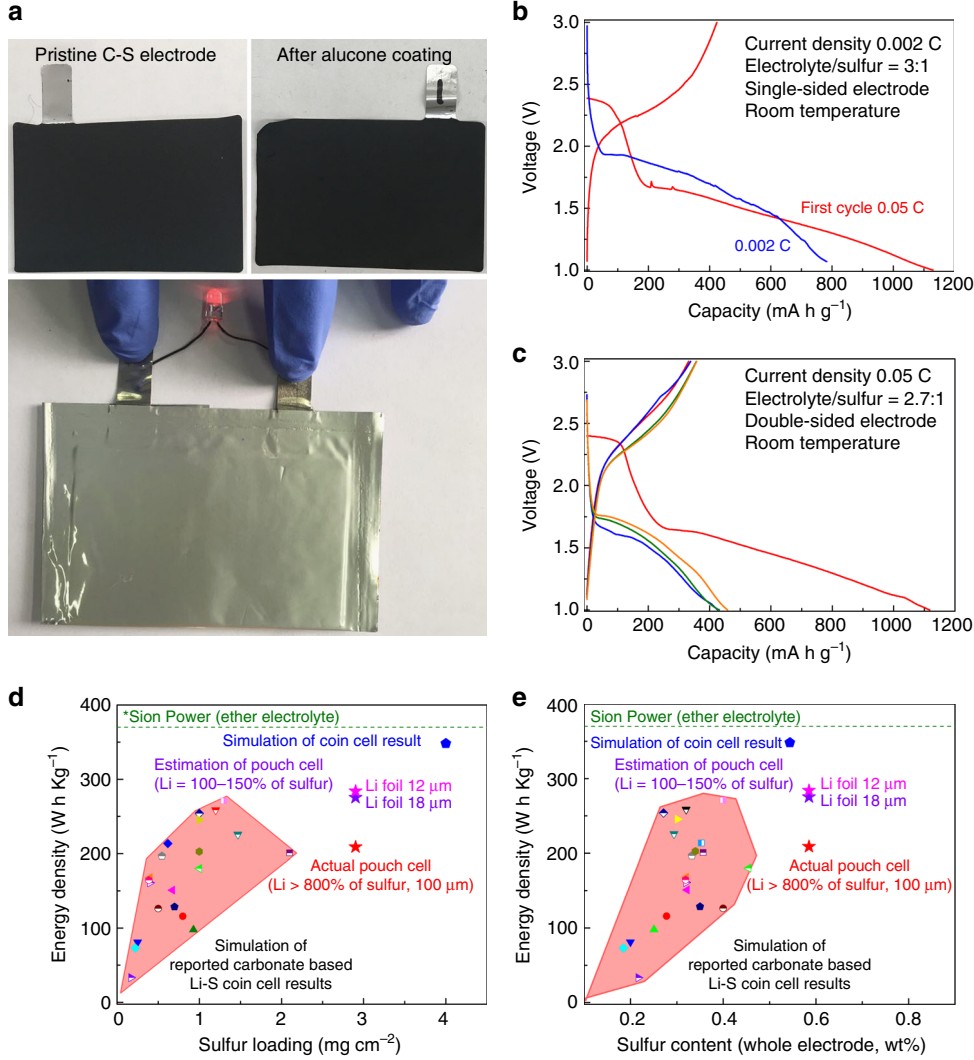

**Fig. 5** Pouch cell test of lithium–sulfur batteries in carbonate-based electrolyte. **a** As-prepared C–S electrodes (before and after alucone coating) and an assembled pouch cell. **b**, **c** Discharge–charge profiles of assembled pouch cells under different test conditions. **d**, **e** Measured and simulated energy density of our work and reported lithium–sulfur coin cells in carbonate electrolyte. The reported energy density of Sion Power lithium–sulfur pouch cell in ether-based electrolyte is labeled as a reference[59]

temperatures. The revealed Li–S redox mechanism and obtained electrochemical performance provide strong evidences that conventional commercial C–S cathodes, employing cyclo-$S_8$ molecules, can be successfully cycled in carbonate-based electrolyte systems. This revelation addresses a long-held ideology that only cathodes with low sulfur loading can be used for carbonate Li–S cells. As shown in Fig. 4g, the electrode in carbonate electrolyte, with a sulfur loading of 1.2 mg cm$^{-2}$, can achieve prolonged and stable cycle life with a capacity of 870 mA h g$^{-1}$ following 300 cycles. Further increasing the sulfur loading to 4.0 mg cm$^{-2}$, a capacity of 705 mA h g$^{-1}$ is observed after 300 cycles. Compared with previous reported literature (Supplementary Table 1), this research demonstrates that high sulfur content and areal loading of sulfur cathodes applied in carbonate Li–S batteries is possible. The excellent performance of high loading sulfur electrodes demonstrates the feasibility of high-energy carbonate-based Li–S cells for practical application.

**Pouch cell test and energy density calculation.** To further investigate the potential practical application of Li–S batteries in carbonate-based electrolyte, pouch cell characterization was

performed. The detailed parameters of pouch cell assembly are shown in the Methods section and Supplementary Table 4. Figure 5a presents the prepared electrodes and an assembled pouch cell. Discharge–charge profiles are shown in Fig. 5b, c. First, it should be noted that the pouch cells with both single-sided and double-sided electrodes are operated successfully in carbonate electrolyte, which, to our knowledge, is the first reported Li–S pouch cell in carbonate electrolyte. The first discharge capacity can reach over 1100 mA h g$^{-1}$ at 0.05 C. The pouch cell exhibits reversibility and can maintain over 780 and 430 mA h g$^{-1}$ at 0.002 and 0.05 C, respectively. The discharge profile from the second cycle displays one plateau, confirming the previously hypothesized solid-phase Li–S reaction for alucone-coated cathodes. Second, it should be noted that there are still many challenges in the operation of pouch cells. A large irreversible capacity can be found from the second cycle. This may be due to (1) incomplete coverage of cyclo-$S_8$ by alucone. This conclusion stems from the observed voltage plateau at 2.3 V in the first cycle by uncovered cyclo-$S_8$. The capacity at this plateau is irreversible and from the second cycle only one discharge plateau is present; (2) limited electrolyte within the pouch cell.

Compared with the coin cells, we controlled the amount of electrolyte added in the pouch cell, which may reduce the Li-ion conductivity of the cell, resulting in decreased capacity; (3) the electrolyte did not have FEC as used in coin cells, which may affect the cycling performance.

The success of the Li–S pouch cell in carbonate electrolyte demonstrates the potential practical application of carbonate-based Li–S batteries. The measured and estimated energy density of our results along with simulated energy density of reported carbonate-based Li–S coin cells are available in Fig. 5d, e. Details regarding the calculation parameters are available in Supplementary Tables 1, 3, 5, and Supplementary Fig. 14. We also added Sion Power's ether-based Li–S pouch cell as a reference[59]. The energy densities are only calculated at the electrode level without any shell or package. The energy density of Sion Power Li–S pouch cell is reported to be >350 W h kg$^{-1}$ from public information source[59]. Herein, we assumed that this energy density value is based on cell level and the package mass ratio in the whole pouch cell is around 5 wt% (estimation based on commercial LIBs). Thereby, the energy density at electrode level of Sion Power is calculated to be around 370 W h kg$^{-1}$, which is marked with a dashed line in the figure for comparison. To estimate the energy density, all reported coin cells in the literature are simulated assuming an $E/S$ ratio of 3:1 and adopting the first cycle discharge capacity, which reflects the highest energy density the batteries can reach. In this study, the measurement of our tested pouch cell energy density can reach over 200 W h kg$^{-1}$ with a 100-μm thick Li foil, which the Li amount is over 840% of sulfur. If a thinner Li foil is used in the future (100–150% of sulfur, corresponding to 12 and 18 μm of Li foils), the energy density can be improved to over 280 W h kg$^{-1}$. Compared to the simulated energy density of previously reported carbonate Li–S coin cells, the pouch cell and coin cell results presented here exhibit a commercially viable energy density. We believe that our present study demonstrates the potential of future Li–S batteries in carbonate-based electrolyte to compete with high-energy-density Li–S cells in ether-based electrolyte.

## Discussion

In summary, this research reveals the underlying mechanism of Li–S batteries in carbonate electrolytes and promotes their practical application. First, a detailed mechanism study is presented to unravel the key factors that govern the reversibility of Li–S batteries in carbonate electrolyte systems. In operando XANES suggests a solid-phase Li–S redox reaction taking place in carbonate electrolyte that involves direct transformation between sulfur and $Li_2S$ without the formation of linear polysulfides. This novel mechanism indicates that the molecular format of sulfur is not a limiting factor for achieving highly reversible Li–S batteries in carbonate electrolyte. The significance of using cyclo-$S_8$ in carbonate electrolyte is to open opportunities for the practical application of Li–S batteries. Second, based on the revealed mechanism, we demonstrate the universality of alucone coating for a variety of sulfur cathodes in carbonate electrolyte. By optimizing the electrolyte and carbon hosts, sulfur cathodes presents promising electrochemical performance in the developed Li–S batteries and are found to be highly reversible across a wide temperature window of −20 to 55 °C. Furthermore, the sulfur cathodes represent a high sulfur content (67 wt% of composites) and loading (4.0 mg cm$^{-2}$). In particular, the research demonstrates that the Li–S pouch cells can reversibly operate in carbonate electrolyte systems, indicating strong potential for practical application. This research sheds light on the use of in operando XANES to reveal intricate reaction mechanisms of Li–S batteries and to streamline the development of high-performance C–S cathodes. We hope the revelation of solid-phase reaction mechanism will trigger increased research interests in high-energy Li–S batteries and promote novel electrode architectures for energy storage systems.

## Methods

**Preparation of C–S composites.** Commercial carbon black powders (KETJEN-BLACK Electro-Conductive carbon black 600 (KJ-EC600), BLACK PEARLS carbon blacks (BP2000, BP1300, BP800), US) were employed as hosts for sulfur. C–S were prepared by mixing carbon black with sulfur powder (99.5%, Sigma-Aldrich) and dried at 80 °C for 12 h to remove moisture. The mixture was then transferred to a sealed steel reactor and heated at 150 °C for 9 h and then 300 °C for 2 h. The obtained C–S composites have a sulfur content of 70 wt%. The KJ-EC600 C–S composites employed in our previous study are used as standard samples in the XANES study and investigation of alternate electrolyte components (in Figs. 2, 3)[7]. BP carbon materials are used to study the influence of carbon hosts, optimize the battery electrochemical performance, and conduct the pouch cell test (Figs. 4, 5).

**Preparation of small sulfur cathode material.** Microporous carbon material is synthesized in a two-step process that incorporates short-chain sulfur within the host. Eight grams of glucose was dissolved in 50 mL of water. The obtained glucose solutions were then transferred into a sealed Teflon-lined autoclave and heated to 120 °C for 6 h. The obtained brown particles were washed with water, filtered, and subsequently dried at 80 °C in air. The obtained carbon precursor was then immersed in KOH solution and dried at 80 °C to produce a carbon–KOH mixture. The dried mixture was then calcinated under argon at 900 °C for 1 h. The obtained microporous carbon powder was then washed and filtrated with water to remove excess KOH. For preparation of C–S composites, microporous carbon was mixed with sulfur powder (99.5 %, Sigma-Aldrich) and dried at 80 °C for 12 h to remove moisture. The mixture was then transferred to a sealed steel reactor and heated to 150 °C for 9 h and then 300 °C for 2 h. The obtained C–S composites have a sulfur content of 30–40 wt%.

**Preparation of C–S electrode.** Electrodes were prepared via slurry casting, with a mass ratio of 8:1:1 between active material, acetylene black, and poly(vinylidene fluoride-co-hexafluoropropylene), respectively. For regular electrode preparation, the slurry was pasted on Al foil with an areal loading of 1.0–1.2 mg cm$^{-2}$. For high areal sulfur loading electrodes, the slurry was pasted on commercial carbon paper to avoid delamination of active material. The as-prepared electrodes were dried at 60 °C over 12 h under vacuum. For the pouch cell electrodes, cathodes were prepared by mixing the active material with sodium carboxymethyl cellulose at a weight ratio of 9:1 in water and coated onto Al foil at room temperature.

**Preparation of alucone coating on C–S electrode.** MLD of alucone was performed in a Gemstar-8 ALD system (Arradiance, USA). Alucone was directly deposited on the C–S electrodes at 120 °C by alternatively introducing trimethyl-laluminium and ethylene glycol. The growth rate for alucone thin film was determined to be around 0.3 nm per cycle. Sulfur loading was found to drop around 3–5 wt% following MLD treatment.

**Preparation of lithium polysulfide solution.** Stoichiometric amounts of $Li_2S$ and sulfur powders was dissolved in dimethoxyethane (DME) solvent and stirred at 80 °C for 10 h.

**Electrochemical characterization.** CR-2032-type coin cells were assembled in an argon-filled glove box. The coin-type cells consisted of a Li foil as an anode, polypropylene membrane (Celgard 2400) as a separator, and a C–S cathode electrode prepared as outlined above. Several electrolyte systems were selected in this research: (1) carbonate-based electrolyte composed of 1 M lithium hexa-fluorophosphate ($LiPF_6$) salt in carbonate solvents with various components and ratios; and (2) ether-based electrolyte composed of 1 M lithium bis(tri-fluoromethanesulfonyl)imide salt in dioxolane (DOL) and DME solution (DOL: DME = 1:1, volume ratio). All batteries were held at OCV for 2 h before testing. Cyclic voltammograms were collected on a versatile multichannel potentiostation 3/Z (VMP3) using a scan rate of 0.1 mV s$^{-1}$ between 1.0 and 3.0 V (vs. Li/Li$^+$). Electrochemical impedance spectroscopy was also performed on the versatile multichannel potentiostat 3/Z (VMP3) by applying an AC voltage of 5 mV amplitude in the 100 kHz–100 mHz frequency range. Charge–discharge characteristics were galvanostatically tested in the range of 1.0–3.0 V (vs. Li/Li$^+$) at room temperature using an Arbin BT-2000 Battery Test equipment. The equilibrium potentials of the cells were obtained by GITT, which consists of a series of current pulses at 100 mA g$^{-1}$ for 1 h, followed by a 5-h relaxation.

**Pouch cell assembly process.** First, the cathode with C/S composite on aluminium was cut into 7.7 × 5 cm$^2$ and dried at 70 °C. Second, 1 × 2 cm$^2$ C/S composite was removed off the cathode with knife to expose the aluminium foil. Third, the exposed aluminium foil and the Al tab were welded together via the ultrasonic spot

welding machine (Shenzhen Kejing Corporation, China). This process was operated under room temperature and 80 W power. The welding frequency was usually 20 KHz. The assembly process of a soft package battery with a single-sided electrode is as follows: (1) two pieces of Al-plastic films were first cut into 8.7 × 6 cm². (2) Three edges of the Al-plastic were sealed with hot pressure machine (Shenzhen Kejing Corporation, China) under 185 °C for 2 s. (3) The lithium foil (8 cm × 5 cm) was covered with the Celgard 2325 membrane and the cathode (welded with tab). (4) A nickel tap was put at the edge of Li foil. (5) The lithium, membrane, and cathode were pressed together with hand and put into the soft package carefully. (6) 5 mL electrolyte (1 M LiPF$_6$, ethylene carbonate: diethyl carbonate = 1:1) was added into the soft package for 6 h and the excess electrolyte was poured out. (7) The left edge of the soft package was sealed with vacuum hot pressure machine (Shenzhen Kejing Corporation, China) under 185 °C. The left edge of soft package should be pressured for several times under 185 °C to ensure that the hot melt adhesives on the tabs were perfectly integrated with the Al-plastic film. The pouch cell assembly process with a double-sided electrode is a little different from the single-sided one, which needs to be cut into 16 × 5 cm² Li foil and adding two separators at each side of the electrode. The Li foil is folded from the middle to wrap the cathode and two separators, while a Ni tap is added into the middle of the folded lithium. The electrolyte adding step and package sealed step are same with the single-sided electrode.

**Physical characterization.** Morphology of C–S electrodes was characterized using a Hitachi S-4800 FE-SEM equipped with an energy dispersive spectrometer. Lithium foil morphology was obtained using a Hitachi 3400N environmental SEM with an accelerating voltage of 5 kV. Batteries were disassembled in an Ar-filled glovebox and cycled Li foils were transferred into the chamber of SEM quickly. Thermogravimetric analysis was carried out in nitrogen from room temperature to 600 °C at a heating rate of 10 °C min$^{-1}$ on an SDT Q600 (TA Instruments). N$_2$ adsorption–desorption isotherms of carbon materials were collected using a Folio Micromeritics TriStar II surface area and pore size analyzer. Raman scattering spectra were obtained using a HORIBA Scientific LabRAM HR Raman spectrometer system equipped with a 532.4 nm laser. All liquid samples were dropped and sealed between two glass slides in Ar-filled glovebox for Raman analysis. RBS measurements were conducted using 1 and 2 MeV He$^+$ beam (Western Tandetron Facility) on the surface to confirm the elements on Li metal. All Li metal anodes were transferred in an Ar-filled glove bag with minimum exposure to air. TOF-SIMS measurements were conducted using 25 keV Bi$^{3+}$ primary ion, rasterized in an area of 200 × 200 μm² with a pixel density of 128 × 128 of sulfur electrodes. The cycled sulfur electrodes were sealed in Ar-filled glovebox and transferred into the vacuum chamber with minimum exposure to air. Synchrotron-based XANES was carried out at the Canadian Light Source (CLS). Sulfur K-edge XANES was collected using fluorescence yield mode on the soft X-ray microcharacterization beamline (SXRMB) at the CLS[30]. For ex situ XANES experiments, C–S electrodes before and after battery testing were prepared in a vacuum environment prior to synchrotron measurements. To avoid sample oxidation, C–S electrodes following discharge–charge test were obtained from coin cells and sealed in a glovebox under Ar and subsequently transferred to the corresponding beamlines for further measurement. For operando synchrotron studies, a custom-designed coin cells with a 5 mm opening on the cathode side was employed. A thin film covers over the opening allowed for beam penetration. Ten-cycle alucone-coated C–S electrodes were employed in the comparison study between carbonate- and ether-based electrolyte systems. The as-prepared short-chain sulfur cathode, outlined above, is also employed in the in operando study. The ether electrolyte used for in operando studies was composed of 1 M lithium perchlorate (LiClO$_4$) solution in DOL:DME with a volume ratio of 1:1. To achieve a good signal-to-noise ratio, an ambient table set-up was used at the SXRMB beamline. The chamber was filled with helium gas to reduce scattering at low energies. Charge–discharge characterizations of operando cells were galvanostatically tested at a current density of 160 mA g$^{-1}$ in the range of 1.0–3.0 V (vs. Li/Li$^+$) at room temperature. The XANES measurements has been done at the shortest time around 9–15 min scans with good quality data at SXRMB beamline.

## Data availability
The data that support the findings of this study are available from the corresponding author upon reasonable request.

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

## Acknowledgements

This research was supported by Natural Sciences and Engineering Research Council of Canada (NSERC), Canada Research Chair Program (CRC), Canada Foundation for Innovation (CFI), Ontario Research Fund, the Canada Light Source at University of Saskatchewan (CLS), Interdisciplinary Development Initiatives (IDI) by Western University, and University of Western Ontario. X.L. thanks the support of Mitacs Elevate Postdoctoral Fellowship.

## Author contributions

X.S. conceived the overall project; X.L. designed the procedure with the help from Q.S. and performed the experiments, data analysis as well as wrote the manuscript; A.L. and Y.Z. performed MLD experiment; X.L., M.B., Q.S., B.W., M.L., Y.H., and T.K.S. performed synchrotron-based XANES characterization; Y.Z. and Q.L. carried out SEM images of Li metal. X.Y., C.L., and H.Z. performed pouch cell test. L.G. performed RBS test. J.L. performed Raman test. C.W., W.X., and R.L. interpreted the results and data analysis. X.L. and Q.S. proposed the reaction mechanisms. All authors read and commented on the manuscript.

## Additional information

**Competing interests:** The authors declare no competing interests.

