## [Peer Review File · Nature Communications]

Reviewers' comments:

Reviewer #1 (Remarks to the Author):

This manuscript reports that Li-S in a carbonate-based electrolyte can be operated irrespective of the molecular format of sulfur by using in-operando XAS. The manuscript provides interesting result and insight. However, I hesitate to recommend the publication

of this manuscript, at least in its present form. There are some aspects of the work that need to be addressed.

1. Many explanations in the manuscript refer to previous paper about the operation of alucone coated Li-S in a carbonate-based electrolyte. This makes the manuscript dependent on previous results rather than independent.

2. Even though the manuscript shows the operation of Li-S in a carbonate-based

electrolyte irrespective of the molecular format of S, they do not explain why and how this can be possible. I think that at least some speculations for this can be provided in the manuscript. Does this behavior strongly depend on the solubility of the reacted products in Li-S into the electrolyte?

3. Considering that S and Li₂S are known as insulating materials, how can the solid phase transformation in charge/discharge process occur?

4. Given that the solid-phase transformation in Li-S can occur, how and why different carbon host can make different electrochemical performance as shown in Fig. 4

Reviewer #2 (Remarks to the Author):

Manuscript Number: NCOMMS-17-34346-T

Title: Eliminating polysulfides via a solid-phase lithium-sulfur transformation: a high energy sulfur cathode in carbonate electrolyte.

I recommend rejection based on the following observations.

The authors have demonstrated a new reaction mechanism in which sulfur is directly converted into Li₂S without soluble polysulfide formation in their material (alucone coated sulfur cathode) in a carbonate-based electrolyte. Although the performance of alucone coated sulfur cathode is good, this was reported in their previous study (Nano Letters, 2016, 16, 3545-3549), so I do not feel that there are

new findings which deserve to be published in Nature Communications. In addition, related to new mechanism, the main claim was very weakly supported. Furthermore, there are several points that the authors should consider.

1) There is no reasonable explanation why the mechanisms in ether-based and carbonate-based electrolytes for alucone coated C-S electrode are different. If eliminating polysulfides is due to confinement effect by small pores (< 2nm), why are two plateaus are still observed in ether-based electrolytes?

2) The authors claim that sulfur directly transfers to Li₂S without polysulfide formation in alucone coated C-S electrode with carbonate-based electrolyte, through a solid-phase transformation, but the supporting data for this claim is very weak. First, this claim is mostly supported by operando XANES results, but the signal to noise is low. For example, the appearance and disappearance of a peak at 2470 eV in figure 2d (S_x²⁻) is unclear. Second, from the Li₂S peak in figure 2e, the authors claimed that the reaction is reversible, but the Li₂S peak that appeared during discharge remained until end of charge. Third, the Li₂S peak in figure 2d looks almost unchanged compared to peak at figure 2e, but it should be different since the final product is Li₂S in both electrolytes.

3) The authors should provide other supporting data for their claims. For example, providing evidence of no color change in the electrolyte during discharge would be direct evidence for elimination/suppression of polysulfide formation.

4) Presenting XRD of both the alucone coated C-S electrode and short-chain sulfur cathode could provide evidence of their difference.

5) Voltage profiles in figures 2a-c and 2d-f are different, especially, the two plateaus are unclear in figure 2d. An explanation should be provided. Also, capacity values in figures 2d-f should be included.

6) In Figure 5e, two peaks are still observed during lithiation even in carbonate electrolyte. If polysulfide formation is eliminated, why are there still two peaks?

Reviewer #3 (Remarks to the Author):

This work discusses the mechanisms of Li-S reactions in carbonate-based electrolyte by coating S/C with alucone. The author mainly claim two things: 1) carbonate is safer than ether and 2) they demonstrate a "high" S loading electrodes in carbonate electrolytes and project the cell energy will be high. However, these two claims are not fully supported by their work.

1. Although carbonate electrolytes, compared to ether-based one, have some advantages such as wider electrochemical window, the cycling of S does not need to go above 3.5 V at which the decomposition of ether begins. More importantly, for lithium metal based batteries (Li-S is a Li metal battery), ether is more benign than carbonate solvents towards lithium metal and SEI generated is much resistance as reported by many literature. DOL/DME indeed has lower vapor pressure and boiling points than carbonate solvents. However, either ether or carbonate, they are both organic solvents and functions as "fuels" once batteries catch fire. From this point of view, there is no big difference between ether or carbonate based electrolytes in terms of safety. Remember, thermal runaway starts from SEI breakdown.

2. The calculation on the cell energy is not accurate. It is not convincing that with 4 mg/cm² S loading, the authors can reach 350 Wh/kg. The amount of minimum electrolyte needed is determined by the cathode architecture. With carbon paper as the host, a large amount of excessive amount of electrolyte (much more than 3 times) will be required significantly reducing the energy. The cell energy calculation is also based on the assumption that they don't need Cu current collector which is not true. The averaged voltage is much lower than those tested in ether-based one which further lowers the energy. It is suggested that the authors preparer a pouch cell by using the method they discussed to see if they can reach the number even at the first discharge.

In addition, Sion Power already demonstrate the same energy density Li-S pouch cell before in ether based electrolytes further confirming that it does no matter which electrolyte is used as long as it works.

3. The carbon host used in this work has high surface area so the authors used a different current collector i.e., carbon paper to make the S/C cathodes to avoid delamination. However, the delamination is a common phenomenon for all electrodes consisting of nano particles. In other words, the authors should address the coating issue of using nano materials instead of using carbon paper as the current collector. It will make the comparison with control experiments not at the same level. Carbon paper cannot be used in pouch cell assembly neither.

4. It seems that in all the experimental results the authors have no control on the E/S ratio which is key to obtain the best performances of Li-S cells. The inconsistence of current collector used also makes the conclusions question mark.

5. The only evidence used to prove that after alucone coating, the S₈ directly converts to L₂S is XANES. however, the lacking of standard reference materials during the transition leads to many uncertainties. If what the authors claimed is true, then why in all-solid-state Li-S, still two clear plateaus are see similar as in liquid cells?

6. The addition of FEC enhances SEI quality on Li. The authors did not discuss what are the interlaces in figure 6S. in carbonate electrolytes, did authors still see Sulfur cross over to the anode side? Or Sulfur redistribution? The main advantage of the proposed mechanism is the direct conversion and avoid the generation of soluble species. Did shuttle still happen though? All these critical questions were not mentioned.

The above work needs to demonstrate more evidence and discuss why alucone coating can prevent continuous electrolyte attack to polysulfide or promote the direct conversion. They also need to prepare a real pouch cell instead of doing simulation to show their claims are correct. Otherwise, the work cannot meet the qualification of NC.

Response to Reviewers' Comments (Manuscript ID NCOMMS-17-34346-T)

We are very grateful for the reviewers' constructive comments and insightful suggestions for improving the quality of this manuscript. In this revision, we have carried out more experiments and added 9 figures and one table to better demonstrate the potential practical application of carbonate electrolyte based Li-S batteries with alucone coating and to give more in-depth analysis on the underlying mechanisms of the solid-phase Li-S reactions. We also cited more related references and added corresponding discussion in the revised manuscript. The yellow highlights indicate what have been added in the manuscript and supporting information. The major new experiments, results, and conclusions include:

1. We have conducted pouch cell test of Li-S battery using alucone coated carbon-sulfur cathodes in carbonate based electrolyte and measured its energy density. To our best knowledge, it is the first public reported carbonate-electrolyte based Li-S pouch cell.
2. We carried out a newly-designed experiment to determine the reactivity of polysulfide versus carbonate electrolyte --- chemically synthesized polysulfide solutions were added to pure carbonate solvent or carbonate electrolytes. The solubility and color changes of polysulfides in carbonate solutions are tracked to understand the possible reaction mechanism. We also assembled the glass-tube cells for directly witnessing the color change of the electrolytes to differ the solution-based or solid-state conversion reaction mechanism in both ether and carbonate electrolytes, respectively.
3. We carried out more in-operando and ex-situ XANES study to further elucidate the mechanisms of solid-phase Li-S reactions and the new data are updated in the revised manuscript.
4. We also carried out various physical and electrochemical characterizations, including Rutherford backscattering spectrometry (RBS), time of flight secondary ion mass spectrometry (TOF-SIMS), X-ray diffraction (XRD), Raman, cyclic voltammetry (CV), and electrochemical impedance spectroscopy (EIS) to give more supporting information to clarify the reaction routes of the Li-S batteries.

Reviewer #1

General comments: This manuscript reports that Li-S in a carbonate based electrolyte can be operated irrespective of the molecular format of sulfur by using in-operando XAS. The manuscript provides interesting result and insight. However, I hesitate to recommend the publication of this manuscript, at least in its present form. There are some aspects of the work that need to be addressed.

Author reply: Thank you very much for your constructive suggestions. We really appreciate the reviewer's suggestions and questions, which are very helpful to improve the quality of this manuscript.

Comment 1: Many explanations in the manuscript refer to previous paper about the operation of alucone coated Li-S in a carbonate based electrolyte. This makes the manuscript dependent on previous results rather than independent.

Author reply: Thanks for the comments. The previous paper explored MLD alucone coating for the sulfur cathode and demonstrates the possibility of its use in carbonate electrolyte. In this paper, we devote more efforts on the detailed mechanistic study and, more importantly, to promote the carbonate based Li-S batteries to practical application. In this revision, particularly, we have performed the practical pouch cell testing, which, to our knowledge, is the first time to report and demonstrate carbonate electrolyte based Li-S pouch cell, as shown in Figure 1. Furthermore, we also carried out more ex-situ and in-operando XANES study, electrochemical characterizations, liquid based chemical experiments, and other physical characterizations (RBS, TOF-SIMS, XRD, and Raman) to give more in-depth analysis on the underlying mechanisms. The newly-added pouch cell test and physical characterizations make this paper more independent and provides a new direction for the development of practical Li-S batteries in the future.

Figure 1. Pouch cell characterization of alucone coated C-S electrodes in carbonate electrolyte. (a) As-prepared C-S electrodes (before and after MLD coating) and the assembled pouch cell. (b) Discharge-charge profiles of assembled pouch cells under different conditions. (c) Calculated energy density of our work and reported Li-S batteries in carbonate electrolyte. The reported energy density of *Sion Power* Li-S pouch cell in ether based electrolyte is labeled as a reference (*Sion Power* news Available online: <https://sionpower.com/2015/airbus-ds-and-sion-power-enter-into-collaborative-agreement>).

***Notes:** The energy density of *Sion Power* Li-S pouch cell is reported to be $> 350 \text{ Wh kg}^{-1}$ from public information source. Herein, we assumed that this energy density value is based on cell-level and the package mass ratio in the whole pouch cell is around 5 wt% (estimated based on commercial LIBs). So the energy density at electrode level of *Sion Power* can be calculated to be

around 370 Wh kg^{-1} , which is marked in dashed line in the figure to compare. [#]In our actual assembled pouch cell, the thinnest Li foil that we can get is $100 \mu\text{m}$ and the amount of Li is over 840% of sulfur. When using thinner Li foils with the thickness of 18 and $12 \mu\text{m}$ which are corresponding to 100% and 150% of sulfur (the same condition in estimation), the energy density will significantly increase.

Figure 1a presents the prepared electrodes and assembled pouch cell. Discharge-charge profiles are shown in Figure 1b and the detailed parameters of pouch cell assembly are shown in Table 1. **Firstly**, it should be noted that the pouch cells with both single-sided and double-sided electrodes are operated successfully in carbonate electrolyte, which, to our knowledge, is the first reported Li-S pouch cell in carbonate electrolyte. The first discharge capacity can reach over 1100 mAh g^{-1} at 0.05 C . The pouch cell demonstrates good cycling reversibility which the 2nd cycle discharge capacity can maintain over 780 and 430 mAh g^{-1} at 0.002 C and 0.05 C , respectively. The discharge profiles from the second cycle performed one plateau, well confirming the solid-phase Li-S reaction. **Secondly**, it should be noted there are still many challenges in the operation of current pouch cells. A big irreversible capacity can be found from the second cycle. This may be due to **(1)** some of cyclo- S_8 was not covered by alucone coating. We can still see the voltage plateau at 2.3 V from the uncovered cyclo- S_8 in the first cycle. The capacity at this plateau is irreversible and from the second cycle only one discharge plateau appeared; **(2)** limited electrolyte added in the pouch cell. Compared with the coin cell, we controlled the amount of electrolyte adding in pouch cell, which may reduce the Li-ion conductivity of the cell, leading to the decrease of capacity; and **(3)** the electrolyte did not have FEC as used in coin-cells, which may affect the cycling performance.

Table 1. Pouch cell assembly parameters

Electrode size	77 mm * 50 mm
Sulfur host	BP2000
Sulfur content	> 65 wt%
Sulfur loading (After MLD)	1.35-1.45 mg cm^{-2} (Single-sided) 2.65-2.95 mg cm^{-2} (Double-sided)
C-S composite: Binder	9:1
Current collector	Al foil
Electrolyte	LiPF_6 (EC:DEC, v:v=1:1)
Electrolyte: Sulfur	< 3:1

Due to time limitations, we will further continue to improve the pouch cell performance in the future. Meanwhile for now, we believe the success demonstration of the Li-S pouch cell in carbonate electrolyte as a first attempt demonstrates the potential practical application of carbonate based Li-S batteries. The measured and estimated energy density of our results and simulated energy density of reported carbonate based Li-S coin cells are shown in Figure 1c. Detailed calculation parameters are shown in supporting information Table S1, S3, S5, and Figure S15. We also added *Sion Power's* ether-based Li-S pouch cell as a reference in the figures. The energy densities are only calculated at the electrode level without any shell or package. To estimate the energy density of pouch cells, all of reported coin-cell data in previous references are simulated assuming the E/S ratio of 3:1 and adopting the first cycle discharge capacity, which reflects the highest energy density the batteries can reach. In this study, the measured actual pouch cell energy density can reach over 200 Wh kg⁻¹ with a thick Li foil of 100 μm, which the Li amount is over 840% of sulfur. If using a thinner Li foil in the future as the estimation condition (100%-150% of sulfur), the energy density can be improved to over 280 Wh kg⁻¹. Compared with the simulated pouch cell energy density of all previous reported carbonate Li-S cells, both our pouch cell and coin cell results exhibits a competitive energy density. We believe that our present study demonstrate the potential of future carbonate-electrolyte based Li-S batteries to compete the high energy density as the Li-S cell in ether based electrolyte.

ACTION: We have added Figure 1 and Table 1 in the manuscript as Figure 5 and supporting information as Table S4, respectively. We also added the follow description in the manuscript:

“To further probe the potential practical application of the developed carbonate electrolyte Li-S batteries, pouch cell characterizations were performed. The detailed parameters of pouch cell assembly are shown in Table S4. Figure 5a presents the prepared electrodes and assembled pouch cell. Discharge-charge profiles are shown in Figure 5b. Firstly, it should be noted that the pouch cells with both single-sided and double-sided electrodes are operated successfully in carbonate electrolyte, which, to our knowledge, is the first reported Li-S pouch cell in carbonate electrolyte. The first discharge capacity can reach over 1100 mAh g⁻¹ at 0.05 C. The pouch cell demonstrates good cycling reversibility which the 2nd cycle discharge capacity can maintain over 780 and 430 mAh g⁻¹ at 0.002 C and 0.05 C, respectively. The discharge profiles from the second cycle performed one plateau, well confirming the solid-phase Li-S reaction. Secondly, it should be noted there are still many challenges in the operation of current pouch cells. A big irreversible capacity can be found from the second cycle. This may be due to (1) some of cyclo-S₈ was not covered by

alucone coating. We can still see the voltage plateau at 2.3 V from the uncovered cyclo-S₈ in the first cycle. The capacity at this plateau is irreversible and from the second cycle only one discharge plateau appeared; (2) limited electrolyte added in the pouch cell. Compared with the coin cell, we controlled the amount of electrolyte adding in pouch cell, which may reduce the Li-ion conductivity of the cell, leading to the decrease of capacity; (3) the electrolyte did not have FEC as used in coin-cells, which may affect the cycling performance.

The success of the Li-S pouch cell in carbonate electrolyte as a first attempt demonstrates the potential practical application of carbonate based Li-S batteries. The measured and estimated energy density of our results and simulated energy density of reported carbonate based Li-S coin cells are shown in Figure 5c. Detailed calculation parameters are shown in supporting information Table S1, S3, S5, Figure S15. We also added *Sion Power's* ether-based Li-S pouch cell as a reference in the figures. The energy densities are only calculated at the electrode level without any shell or package. To estimate the energy density of pouch cells, all of reported coin-cell data in previous references are simulated assuming the E/S ratio of 3:1 and adopting the first cycle discharge capacity, which reflects the highest energy density the batteries can reach. In this study, the measured actual pouch cell energy density can reach over 200 Wh kg⁻¹ with a thick Li foil of 100 μm, which the Li amount is over 840% of sulfur. If using a thinner Li foil in the future as the estimation condition (100%-150% of sulfur), the energy density can be improved to over 280 Wh kg⁻¹. Compared with the simulated pouch cell energy density of all previous reported carbonate Li-S cells, both our pouch cell and coin cell results exhibit a competitive energy density. We believe that our present study demonstrates the potential of future carbonate-electrolyte based Li-S batteries to compete the high energy density as the Li-S cell in ether based electrolyte.”

Comment 2: Even though the manuscript shows the operation of Li-S in a carbonate based electrolyte irrespective of the molecular format of S, they do not explain why and how this can be possible. I think that at least some speculations for this can be provided in the manuscript. Does this behavior strongly depend on the solubility of the reacted products in Li-S into the electrolyte?

Author reply: That's an excellent question! We have designed a liquid based chemical experiments to probe the compatibility (solubility) of polysulfides in carbonate electrolyte and propose the possible mechanism of irreversible Li-S reaction in carbonate electrolyte.

To illustrate the solubility of polysulfides in carbonate electrolytes, we prepared the solutions with 1M LiPF₆ within DEC and EC solvent (Solution A); pure DEC solvent (Solution B); and two different mixing ratio with solution A and B (A: B= 2:1 and 1:1) to obtain the different concentration of LiPF₆ salt in the carbonate solvents, as shown in Figure 2a. With 5-7 drops of polysulfide solution (Li₂S_x, x≤6, DME solvent), the four solutions have different changes, as shown in Figure 2b-d. For pure DEC solvents (solution B), the polysulfide solution can easily mix in it and no precipitate was form in the solution. For the carbonate solutions with LiPF₆ salt, the polysulfide solution is hard to mix in it. Especially the carbonate solution with a high concentration of LiPF₆ (1M, solution A), a large amount of yellow precipitate is formed in a short time (Figure 2b-d). Raman characterization was carried out to probe the precipitate materials, as shown in Figure 2e, which indicates the yellow material formed in the solutions is elemental sulfur. These experiments lead us to the following conclusion: lithium polysulfides are highly instable when co-existed with LiPF₆ based carbonate electrolyte in sharp contrast to merely carbonate solvent, where side reaction occurs with the formation of elemental sulfur as a product. These new results are fascinating and in contrast to the previous assumption that the side chemical reactions are simply between carbonate solvent molecular and polysulfide.

Figure 2. Solubility evolution of polysulfides in carbonate based solutions. (a-d) Evolution of different carbonate based liquid with the added drops of polysulfides solution. (e) Raman spectra of the carbonate solutions with polysulfides.

Based on the experiments, we propose the reaction mechanism as follows: when cyclo-S₈ discharge in carbonate electrolyte, the formed polysulfides will be quickly decomposed by carbonate electrolyte and partially transformed to elemental sulfur. This side-reaction consumes the active polysulfides and the re-formed sulfur may diffuse in electrolyte, block the porous carbon, and cover the electrode. As a result, these side products including elemental sulfur precipitates without

conductive agent are hard to be further utilized in the electrochemical reaction and also blocks the reaction of remaining sulfur inside of the electrode, leading to an irreversible Li-S battery. On the other hand, the conformal alucone coating on C-S composites effectively avoids this side reaction by shielding the carbonate electrolyte following the reaction route that the active sulfur encapsulated in conductive carbon to undergo a solid-phase Li-S redox reaction, leading to the reversible electrochemical reaction in carbonate electrolyte.

It should be noted that many detailed mechanisms of the reaction between polysulfides with carbonate electrolyte is still to be discovered and the obtained phenomenon that polysulfides may have difficulty to exist with high concentration LiPF_6 salt in carbonate solvents is also different from the previous speculation that polysulfides reacted with carbonate solvents. We will do more research to further probe the mechanism in the future.

ACTION:

Figure 2 is added in supporting information as Figure S12 and the description of this manuscript have added as follows:

In manuscript:

“The mechanism of cyclo- S_8 based cathode operation in carbonate based electrolyte may be related to the compatibility of polysulfides. As shown in Figure S12, we designed a liquid based chemical reaction to probe the solubility of polysulfides in carbonate solutions. Detailed experiment processes are shown in the supporting information. The experiment led us to draw the following conclusion: lithium polysulfides are highly instable when co-exist with LiPF_6 based carbonate electrolyte, which results in the decomposition and partially conversion of polysulfides to elemental sulfur in the carbonate electrolyte. The conformal alucone coating on C-S composites, on the other hand, effectively avoids the side-reaction to form a large amount of sulfur precipitates in the electrolyte and therefore forces the sulfur encapsulated in conductive carbon to undergo a solid-phase Li-S redox reaction, leading to the reversible electrochemical reaction in carbonate-based electrolyte.”

In supporting information:

“To illustrate the solubility of polysulfides in carbonate electrolytes, we prepared the solutions with 1M LiPF_6 in DEC and EC solvent (Solution A); pure DEC solvent (Solution B); and two different mixing ratios with solution A and B (A: B= 2:1 and 1:1) to obtain the different concentration of

LiPF₆ salt in the carbonate solvents, as shown in Figure S12a. With 5-7 drops of polysulfide solution (Li₂S_x, x≤6, DME solvent), the four solutions experienced different changes, as shown in Figure S12b-d. For pure DEC solvents (solution B), the polysulfide solution easily mixed in it and no precipitate formed. For the carbonate solutions mixed with LiPF₆ salt, the polysulfide solution is hard to mix with it. In the carbonate solution with a high concentration of LiPF₆ (1M, solution A), a large amount of yellow precipitate was formed in a short time (Figure S12b-d). Raman characterization was carried out to probe the precipitate materials, as shown in Figure S12e, which indicates that the yellow material formed in the solutions is elemental sulfur. These experiments led us to the following conclusion: lithium polysulfides are highly instable when co-existed with LiPF₆ based carbonate electrolyte in sharp contrast to merely carbonate solvent, where side reaction occurs with the formation of elemental sulfur as a product.”

Comment 3: Considering that S and Li₂S are known as insulating materials, how can the solid phase transformation in charge/discharge process occur?

Author reply: Thank you for the question. As you mentioned, S and Li₂S are insulating materials. Thus sulfur cathode for Li-S batteries are often used in the form of sulfur-carbon composite where sulfur are refined into the pores of the carbon host. These S/C cathodes are in fact the most used sulfur cathode for both ether and carbonate electrolyte-based Li-S batteries. Therefore, to increase the conductivity of electrode, we encapsulated sulfur in highly conductive porous carbon materials, which helps sulfur to transfer electrons and complete the Li-S electrochemical reaction. We also conducted EIS test to further demonstrate it. As shown in Figure 3, after twenty discharge-charge cycles in carbonate electrolyte, the battery using alucone coated C-S electrode still maintained very small surface charge transfer resistance, which indicates good conductivity of as-prepared sulfur cathode. However, for the battery with bare C-S electrode, due to the side-reaction mentioned before that much sulfur covered on the electrode (Comment 2), the resistance of the battery was very large after one discharge-charge cycle. Therefore, the introduced carbon hosts for sulfur cathodes provided effective electron conductive network for the Li-S redox reaction.

Figure 3. EIS plots of the Li-S batteries with different sulfur cathodes after cycling.

On the other hand, the solid-phase transformation of the Li-S reaction have also been reported in many recent reports regarding all-solid-state Li-S batteries with inorganic thiophosphate solid-state electrolyte (*Nano Lett.*, 2016, 7, 4521; *Adv. Energy Mater.*, 2017, 7, 1602923; *J. Mater. Chem. A*, 2017, 5, 6310). The reaction kinetic in solid-state batteries is absolutely slower than liquid-state batteries. However, accompanied with conductive carbon and highly ionic conductive solid-state electrolytes, the insulated sulfur cathodes or Li₂S cathodes can also discharge/charge in solid-phase with promising cycling performance. Thereby with the conductive agent, sulfur and Li₂S undergoing solid-phase transformation is of course possible in both liquid-state and solid-state Li-S batteries.

ACTION: We have added Figure 3 and following description in supporting information as Figure S13: “Figure S13 demonstrates the EIS spectra of the Li-S batteries with different sulfur cathodes cycled in carbonate electrolyte. After twenty discharge-charge cycles in carbonate electrolyte, the battery using alucone coated C-S electrode still maintained very small surface charge transfer resistance, which indicates good conductivity of as-prepared sulfur cathode. However, the battery with bare C-S electrode experienced very large resistance after one discharge-charge cycle.”

We also cited some reported all-solid-state Li-S batteries with inorganic solid-state electrolyte to demonstrate the solid-phase Li-S reaction:

42. Yao X, *et al.* High-Performance All-Solid-State Lithium-Sulfur Batteries Enabled by Amorphous Sulfur-Coated Reduced Graphene Oxide Cathodes. *Adv. Energy Mater.* **7**, 1602923 (2017).
43. Han F, *et al.* High-Performance All-Solid-State Lithium-Sulfur Battery Enabled by a Mixed-Conductive Li₂S Nanocomposite. *Nano Lett.* **16**, 4521-4527 (2016).
44. Xu R-c, Xia X-h, Li S-h, Zhang S-z, Wang X-l, Tu J-p. All-solid-state lithium-sulfur batteries based on a newly designed Li₇P_{2.9}Mn_{0.1}S_{10.7}I_{0.3} superionic conductor. *J Mater. Chem. A* **5**, 6310-6317 (2017).

Comment 4: Given that the solid phase transformation in Li-S can occur, how and why different carbon host can make different electrochemical performance as shown in Figure 4

Author reply: That's a very good question. The properties of carbon hosts for sulfur cathodes that follow the solid-liquid dual-phase reaction has been widely investigated, for example, the pore size, surface area, doping effect, and hydrophilic effect (*Acc. Chem. Res.*, 2012, 45, 1759; *Angew. Chem. Int. Ed.*, 2016, 55, 12990; *Adv. Mater.* 2017, 29, 1606823). For the new solid-phase Li-S reaction, the properties of carbon hosts may also impact the performance of the Li-S batteries.

Figure 4. (a) Comparison of the three carbon hosts and the corresponding CV cathodic peaks of Li-S batteries. (b) Schematic figure of the sulfur distribution in carbon hosts with various porous structures.

In the paper, we have compared the performance of the sulfur cathodes with three types of carbon hosts and demonstrated that the carbon host with the smallest pore size (< avg. 4 nm) and highest surface area (> 1300 m² g⁻¹) improves the performance of sulfur cathode in carbonate electrolyte the most. Figure 4a compared the cathodic peaks of the Li-S batteries with different carbon hosts in CV profiles (Supporting information Figure S10). The sulfur cathode with smaller porous structure presents higher cathodic peaks in the CV test, indicating a reduced potential polarization of the Li-S

battery. This phenomenon may be related to the distribution of sulfur and electronic conductive network in the carbon host. Figure 4b simulates the distribution of sulfur in porous carbon hosts. With the same amount of sulfur in the C-S composite, the smaller porous structure encapsulates sulfur widely scattered while the larger pore aggregates more sulfur together. Due to the insulation nature of sulfur, the electronic conductivity of cathode is very crucial for the solid-phase Li-S reaction. Considering the limited quantum electron tunneling length, it may be implied that the sulfur in a larger hole with the diameter above 10 nm should more unlikely to have a good electrochemical performance, which is in good consistency with the relationship between the capacity and pore distribution of the three carbon host materials (Figure 4 in manuscript). The widely dispersed sulfur increases the contact area with carbon and improves the conductive network for the sulfur cathode. Therefore, the carbon host with high surface area and confined porous structure has demonstrated the best performance of sulfur cathodes in solid-phase Li-S reaction.

ACTION: Figure 4b is as Figure S11 added in supporting information. We also replace the Figure S10 as following and added description in the manuscript:

In manuscript:

“The differences observed in cycling performance between three sulfur electrodes are proposed to originate from the nanostructured carbon hosts. Compared to the other carbon hosts, BP2000 has the highest surface area with confined porous architecture. This unique structure allows for highly dispersed sulfur distribution throughout the host material and improved electronic conductivity of sulfur in the solid-phase Li-S reaction.”

“Cyclic voltammograms of the three alucone coated C-S cathodes are presented in Figure 4d and Figure S10. All coated cathodes demonstrated reversible Li-S redox electrochemical reaction in carbonate electrolytes, with a single pair of cathodic and anodic peaks at around 1.75 V and 2.3 V, respectively. Compared to the other two carbon hosts as shown in Figure S10, the sulfur cathode with BP2000 exhibited sharp and highly reversible peaks. Furthermore, the cathodic peak potential of the sulfur cathode with BP2000 was also higher than the others, indicating smaller potential polarization and higher electrochemical reaction activity of the Li-S battery.”

Figure S10. Effect of porous structure on the performance of Li-S batteries in carbonate electrolyte.

(a) Pore size distribution of the three carbon hosts. (b-d) CV profiles of the Li-S batteries with different carbon hosts. (e) Summary of the three carbon hosts in terms of surface area, average pore size, and corresponding cathodic peaks of sulfur cathodes. (f) CV profiles of Li-S batteries in ether based electrolyte.

Reviewer #2

General comments: The authors have demonstrated a new reaction mechanism in which sulfur is directly converted into Li_2S without soluble polysulfide formation in their material (alucone coated sulfur cathode) in a carbonate based electrolyte. Although the performance of alucone coated sulfur cathode is good, this was reported in their previous study (Nano Letters, 2016, 16, 35453549), so I do not feel that there are new findings which deserve to be published in Nature Communications. In addition, related to new mechanism, the main claim was very weakly supported. Furthermore, there are several points that the authors should consider

Author reply: Thank you very much for the question and suggestions. They are very important to this paper. Actually, the last paper explored MLD alucone coated sulfur cathode and its application in carbonate electrolyte. In this paper, we for the first time demonstrated the solid-phase Li-S transformation via in-operando XANES and, more importantly, to demonstrate the possibility of the carbonate based Li-S batteries to practical application. In this revision, particularly, we have performed the practical pouch cell testing, which, to our knowledge, is the first time to report and demonstrate carbonate electrolyte based Li-S pouch cell, as shown in Figure 1. Furthermore, we also carried out more ex-situ and in-operando XANES study, electrochemical characterizations, liquid based chemical experiments, and other physical characterizations (RBS, TOF-SIMS, XRD, and Raman) to give more in-depth analysis on the underlying mechanisms. The newly added pouch cell test and physical characterizations make this paper more independent and provides a new direction for the development of practical Li-S batteries in the future.

Figure 1. Pouch cell characterization of alucone coated C-S electrodes in carbonate electrolyte. (a) As-prepared C-S electrodes (before and after MLD coating) and the assembled pouch cell. (b) Discharge-charge profiles of assembled pouch cells under different conditions. (c) Calculated energy density of our work and reported Li-S batteries in carbonate electrolyte. The reported energy density of *Sion Power* Li-S pouch cell in ether based electrolyte is labeled as a reference (*Sion Power* news Available online: <https://sionpower.com/2015/airbus-ds-and-sion-power-enter-into-collaborative-agreement>).

***Notes:** The energy density of *Sion Power* Li-S pouch cell is reported to be $> 350 \text{ Wh kg}^{-1}$ from public information source. Herein, we assumed that this energy density value is based on cell-level and the package mass ratio in the whole pouch cell is around 5 wt% (estimated based on commercial LIBs). So the energy density at electrode level of *Sion Power* can be calculated to be

around 370 Wh kg^{-1} , which is marked in dashed line in the figure to compare. [#]In our actual assembled pouch cell, the thinnest Li foil that we can get is $100 \mu\text{m}$ and the amount of Li is over 840% of sulfur. When using thinner Li foils with the thickness of 18 and $12 \mu\text{m}$ which are corresponding to 100% and 150% of sulfur (the same condition in estimation), the energy density will significantly increase.

Figure 1a presents the prepared electrodes and assembled pouch cell. Discharge-charge profiles are shown in Figure 1b and the detailed parameters of pouch cell assembly are shown in Table 1. **Firstly**, it should be noted that the pouch cells with both single-sided and double-sided electrodes are operated successfully in carbonate electrolyte, which, to our knowledge, is the first reported Li-S pouch cell in carbonate electrolyte. The first discharge capacity can reach over 1100 mAh g^{-1} at 0.05 C. The pouch cell demonstrates good cycling reversibility which the 2nd cycle discharge capacity can maintain over 780 and 430 mAh g^{-1} at 0.002 C and 0.05 C, respectively. The discharge profiles from the second cycle performed one plateau, well confirming the solid-phase Li-S reaction. **Secondly**, it should be noted there are still many challenges in the operation of current pouch cells. A big irreversible capacity can be found from the second cycle. This may be due to (1) some of cyclo-S₈ was not covered by alucone coating. We can still see the voltage plateau at 2.3 V from the uncovered cyclo-S₈ in the first cycle. The capacity at this plateau is irreversible and from the second cycle only one discharge plateau appeared; (2) limited electrolyte added in the pouch cell. Compared with the coin cell, we controlled the amount of electrolyte adding in pouch cell, which may reduce the Li-ion conductivity of the cell, leading to the decrease of capacity; and (3) the electrolyte did not have FEC as used in coin-cells, which may affect the cycling performance.

Table 1. Pouch cell assembly parameters

Electrode size	77 mm * 50 mm
Sulfur host	BP2000
Sulfur content	> 65 wt%
Sulfur loading (After MLD)	1.35-1.45 mg cm^{-2} (Single-sided) 2.65-2.95 mg cm^{-2} (Double-sided)
C-S composite: Binder	9:1
Current collector	Al foil
Electrolyte	LiPF ₆ (EC:DEC, v:v=1:1)
Electrolyte: Sulfur	< 3:1

Due to time limitations, we will further continue to improve the pouch cell performance in the future. Meanwhile for now, we believe the success demonstration of the Li-S pouch cell in carbonate electrolyte as a first attempt demonstrates the potential practical application of carbonate based Li-S batteries. The measured and estimated energy density of our results and simulated energy density of reported carbonate based Li-S coin cells are shown in Figure 1c. Detailed calculation parameters are shown in supporting information Table S1, S3, S5, and Figure S15. We also added *Sion Power's* ether-based Li-S pouch cell as a reference in the figures. The energy densities are only calculated at the electrode level without any shell or package. To estimate the energy density of pouch cells, all of reported coin-cell data in previous references are simulated assuming the E/S ratio of 3:1 and adopting the first cycle discharge capacity, which reflects the highest energy density the batteries can reach. In this study, the measured actual pouch cell energy density can reach over 200 Wh kg⁻¹ with a thick Li foil of 100 μm, which the Li amount is over 840% of sulfur. If using a thinner Li foil in the future as the estimation condition (100%-150% of sulfur), the energy density can be improved to over 280 Wh kg⁻¹. Compared with the simulated pouch cell energy density of all previous reported carbonate Li-S cells, both our pouch cell and coin cell results exhibit a competitive energy density. We believe that our present study demonstrates the potential of future carbonate-electrolyte based Li-S batteries to compete the high energy density as the Li-S cell in ether based electrolyte.

ACTION: We have added Figure 1 and Table 1 in the manuscript as Figure 5 and supporting information as Table S4, respectively. We also added the following description in the manuscript:

“To further probe the potential practical application of the developed carbonate electrolyte Li-S batteries, pouch cell characterizations were performed. The detailed parameters of pouch cell assembly are shown in Table S4. Figure 5a presents the prepared electrodes and assembled pouch cell. Discharge-charge profiles are shown in Figure 5b. Firstly, it should be noted that the pouch cells with both single-sided and double-sided electrodes are operated successfully in carbonate electrolyte, which, to our knowledge, is the first reported Li-S pouch cell in carbonate electrolyte. The first discharge capacity can reach over 1100 mAh g⁻¹ at 0.05 C. The pouch cell demonstrates good cycling reversibility which the 2nd cycle discharge capacity can maintain over 780 and 430 mAh g⁻¹ at 0.002 C and 0.05 C, respectively. The discharge profiles from the second cycle performed one plateau, well confirming the solid-phase Li-S reaction. Secondly, it should be noted there are still many challenges in the operation of current pouch cells. A big irreversible capacity can be found from the second cycle. This may be due to (1) some of cyclo-S₈ was not covered by

alucone coating. We can still see the voltage plateau at 2.3 V from the uncovered cyclo-S₈ in the first cycle. The capacity at this plateau is irreversible and from the second cycle only one discharge plateau appeared; (2) limited electrolyte added in the pouch cell. Compared with the coin cell, we controlled the amount of electrolyte adding in pouch cell, which may reduce the Li-ion conductivity of the cell, leading to the decrease of capacity; (3) the electrolyte did not have FEC as used in coin-cells, which may affect the cycling performance.

The success of the Li-S pouch cell in carbonate electrolyte as a first attempt demonstrates the potential practical application of carbonate based Li-S batteries. The measured and estimated energy density of our results and simulated energy density of reported carbonate based Li-S coin cells are shown in Figure 5c. Detailed calculation parameters are shown in supporting information Table S1, S3, S5, Figure S15. We also added *Sion Power's* ether-based Li-S pouch cell as a reference in the figures. The energy densities are only calculated at the electrode level without any shell or package. To estimate the energy density of pouch cells, all of reported coin-cell data in previous references are simulated assuming the E/S ratio of 3:1 and adopting the first cycle discharge capacity, which reflects the highest energy density the batteries can reach. In this study, the measured actual pouch cell energy density can reach over 200 Wh kg⁻¹ with a thick Li foil of 100 μm, which the Li amount is over 840% of sulfur. If using a thinner Li foil in the future as the estimation condition (100%-150% of sulfur), the energy density can be improved to over 280 Wh kg⁻¹. Compared with the simulated pouch cell energy density of all previous reported carbonate Li-S cells, both our pouch cell and coin cell results exhibit a competitive energy density. We believe that our present study demonstrates the potential of future carbonate-electrolyte based Li-S batteries to compete the high energy density as the Li-S cell in ether based electrolyte.”

Comment 1: There is no reasonable explanation why the mechanisms in ether based and carbonate based electrolytes for alucone coated CS electrode are different. If eliminating polysulfides is due to confinement effect by small pores (< 2 nm), why are two plateaus still observed in ether based electrolytes?

Author reply: Thank you for the question. Actually the polysulfides elimination of alucone coated C-S electrode is not due to the confinement effect by small pores but rather by the solid-phase Li-S transformation. The confinement effect by microporous structure (< 2 nm) for sulfur cathode are

mostly called *small sulfur molecule*, which breaks the cyclo-S₈ molecule to the short-chain S_x molecule (x<4) (*J. Am. Chem. Soc.* 134, 45, 18510). Due the short-chain nature, the small sulfur molecule in both ether and carbonate electrolyte only present one discharge plateau.

In our study, Alucone coated C-S cathodes, however, is totally different from the small sulfur molecule. Firstly, we still employ cyclo-S₈ molecule as active material. The three carbon hosts we employed have various porous structure (Pore size: BP800 Avg. 25 nm; BP1300 Avg. 10 nm; BP2000 Avg. 4 nm, respectively), which circumvents the formation of small sulfur molecule mentioned before. Therefore, as-prepared alucone coated C-S electrodes still employed cyclo-S₈ molecule in the electrodes, which is also the reason alucone C-S electrode can reach high sulfur loading and content.

Secondly, the reasons of why the alucone coated cyclo-S₈ electrode is able to operate in carbonate electrolyte, which is different from the normal cyclo-S₈ cathodes may be due to the compatibility of polysulfides in carbonate electrolytes. To demonstrate this point, we have design a simple chemical experiment as shown in Figure 2. We prepared the solutions with 1M LiPF₆ within DEC and EC solvent (Solution A); pure DEC solvent (Solution B); and two different mixing ratio with solution A and B (A: B= 2:1 and 1:1) to obtain the different concentration of LiPF₆ salt in the carbonate solvents, as shown in Figure 2a. With 5-7 drops of polysulfide solution (Li₂S_x, x≤6, DME solvent), the four solutions have different changes, as shown in Figure 2b-d. For pure DEC solvents (solution B), the polysulfide solution can easily mix in it and no precipitate was form in the solution. For the carbonate solutions with LiPF₆ salt, the polysulfide solution is hard to mix in it. Especially the carbonate solution with a high concentration of LiPF₆ (1M, solution A), a large amount of yellow precipitate is formed in a short time (Figure 2b-d). Raman characterization was carried out to probe the precipitate materials, as shown in Figure 2e, which indicates the yellow material formed in the solutions is elemental sulfur. These experiments lead us to the following conclusion: lithium polysulfides are highly instable when co-existed with LiPF₆ based carbonate electrolyte in sharp contrast to merely carbonate solvent, where side reaction occurs with the formation of elemental sulfur as a product. These new results are fascinating and in contrast to the previous assumption that the side chemical reactions are simply between carbonate solvent molecular and polysulfide.

Figure 2. Solubility evaluations of polysulfides in carbonate based solutions. (a-d) Evolution of different carbonate based liquid with the added drops of polysulfides solution. (e) Raman spectra of the carbonate solutions with polysulfides.

Based on the experiments, we propose the reaction mechanism as follows: when cyclo- S_8 discharge in carbonate electrolyte, the formed polysulfides will be quickly decomposed by carbonate electrolyte and partially transformed to elemental sulfur. This side-reaction consumes the active polysulfides and the re-formed sulfur may diffuse in electrolyte, block the porous carbon, and cover the electrode. As a result, these side products including elemental sulfur precipitates without conductive agent are hard to be further utilized in the electrochemical reaction and also blocks the reaction of remaining sulfur inside of the electrode, leading to an irreversible Li-S battery. On the other hand, the conformal alucone coating on C-S composites effectively avoids this side reaction by shielding the carbonate electrolyte following the reaction route that the active sulfur encapsulated in conductive carbon to undergo a solid-phase Li-S redox reaction, leading to the reversible electrochemical reaction in carbonate electrolyte.

Figure 3. Chemical structure of alucone thin film, DME solvent and PEO polymer. (*Adv. Funct. Mater.*, 23, 2013, 532.)

Thirdly, as the reviewer mentioned, the reason of why the alucone coated cyclo-S₈ electrode in the ether and carbonate electrolytes undergo different reaction manners may be due to the properties of alucone coating itself, as shown in Figure 3. The chemical structure of alucone thin film, PEO polymer, and DME solvent (the most popular ether solvent in electrolyte) actually are very similar. We predict that there is a chance after long time soaking in ether based electrolyte (LiTFSI in DME/DOL), the alucone coating layer is partially turned to gel thin film. The gel alucone thin film allows polysulfides to diffuse into it and therefore the sulfur cathodes still undergo two potential plateaus (solid-liquid dual phase reaction), which is similar to the reported polymer solid-state PEO gel based Li-S batteries. (*J. Phys. Chem. Lett.* 2017, 8, 3473; *J. Mater. Chem. A*, 2017, 5, 12934; *J. Phys. Chem. Lett.* 2017, 8, 1956). The following electrolyte color measurement also demonstrated the polysulfides dissolution from alucone coated C-S electrode in ether based electrolyte, as shown in Figure 5. On the other hand, in carbonate solvents, the alucone thin film will not turn to gel-polymer, leading to solid-phase Li-S batteries. Therefore, we proposed that the different compatibility of polysulfides and the different physical properties of alucone thin film in carbonate and ether based electrolytes result in two types of electrochemical reaction manners of alucone coated C-S electrodes in the two electrolytes.

The detailed reaction mechanisms of carbonate electrolyte and polysulfides and the physical properties of alucone thin film in ether electrolyte mentioned above actually are still not fully understood. The herein obtained phenomenon that polysulfides may be difficult to co-exist with LiPF₆ in carbonate solvents is also different from the previous assumptions that polysulfides react

with carbonate solvents. We expect to devote more research to further probe the mechanisms in the future.

ACTION: Figure 2 is added in supporting information as Figure S12 and the description of this manuscript have added as follows:

In manuscript:

“The mechanism of cyclo-S₈ based cathode operation in carbonate based electrolyte may be related to the compatibility of polysulfides. As shown in Figure S12, we designed a liquid based chemical reaction to probe the solubility of polysulfides in carbonate solutions. Detailed experiment processes are shown in the supporting information. The experiment led us to draw the following conclusion: lithium polysulfides are highly instable when co-exist with LiPF₆ based carbonate electrolyte, which results in the decomposition and partially conversion of polysulfides to elemental sulfur in the carbonate electrolyte. The conformal alucone coating on C-S composites, on the other hand, effectively avoids the side-reaction to form a large amount of sulfur precipitates in the electrolyte and therefore forces the sulfur encapsulated in conductive carbon to undergo a solid-phase Li-S redox reaction, leading to the reversible electrochemical reaction in carbonate-based electrolyte.”

In supporting information:

“To illustrate the solubility of polysulfides in carbonate electrolytes, we prepared the solutions with 1M LiPF₆ in DEC and EC solvent (Solution A); pure DEC solvent (Solution B); and two different mixing ratios with solution A and B (A: B= 2:1 and 1:1) to obtain the different concentration of LiPF₆ salt in the carbonate solvents, as shown in Figure S12a. With 5-7 drops of polysulfide solution (Li₂S_x, x≤6, DME solvent), the four solutions experienced different changes, as shown in Figure S12b-d. For pure DEC solvents (solution B), the polysulfide solution easily mixed in it and no precipitate formed. For the carbonate solutions mixed with LiPF₆ salt, the polysulfide solution is hard to mix with it. In the carbonate solution with a high concentration of LiPF₆ (1M, solution A), a large amount of yellow precipitate was formed in a short time (Figure S12b-d). Raman characterization was carried out to probe the precipitate materials, as shown in Figure S12e, which indicates that the yellow material formed in the solutions is elemental sulfur. These experiments led us to the following conclusion: lithium polysulfides are highly instable when co-existed with LiPF₆

based carbonate electrolyte in sharp contrast to merely carbonate solvent, where side reaction occurs with the formation of elemental sulfur as a product.”

Comment 2 and 5: The authors claim that sulfur directly transfers to Li_2S without polysulfide formation in alucone coated CS electrode with carbonate based electrolyte, through a solid phase transformation, but the supporting data for this claim is very weak. First, this claim is mostly supported by operando XANES results, but the signal to noise is low. (1) The appearance and disappearance of a peak at 2470 eV in figure 2d (S_x) is unclear. The Li_2S peak in figure 2d looks almost unchanged compared to peak at figure 2e, but it should be different since the final product is Li_2S in both electrolytes. (2) From the Li_2S peak in figure 2e, the authors claimed that the reaction is reversible, but the Li_2S peak that appeared during discharge remained until end of charge. (3) Voltage profiles in figures 2ac and 2df are different, especially, the two plateaus are unclear in figure 2d. An explanation should be provided. Also, capacity values in figures 2df should be included.

Author reply: Thank you for the questions and suggestions on the in-situ results, which are very helpful to improve the quality of paper.

Firstly, to address the reviewer’s questions on the Figure 2d in the manuscript (alucone C-S electrode in ether electrolyte), **we did the in-operando XANES again for better spectra quality**. This time we reduced the amount of electrolyte and fortunately got good quality of spectra to clearly observe the evolution of the sulfur cathode in ether based electrolyte, as shown in Figure 4. The spectra has demonstrated a feature at around 2470 eV belonging to chain polysulfides S_x^{2-} . The intensity of this feature was changing along with the discharge-charge process, indicating the redox reaction of chain polysulfides. We also added some new references to make a better comparison of the operando result.

Secondly, the reviewer raised the doubt on the reversibility of alucone coated sulfur cathode in carbonate electrolyte. Actually, the first cycle coulombic efficiency of alucone coated C-S cathode is around 90%, which indicates some of the Li_2S is not reversed back to sulfur. Therefore, it is normal that at the end of first charging process in the operando XANES study we can still see the Li_2S peak in the XANES spectrum.

Figure 4. (2d-f in manuscript) In-operando XANES study of (d) Alucone coated C-S electrode in ether-based electrolyte, (e) alucone coated C-S electrode in carbonate-based electrolyte, and (f) as-prepared small sulfur molecule cathode in carbonate-based electrolyte.

Thirdly, the reviewer is concerned about the discharge-charge profiles in operando study, especially the ones in ether-based electrolyte (Figure 2d in manuscript). As shown in Figure 4, the newly obtained discharge-charge profiles have clear two discharge plateaus, corresponding to the solid-liquid dual-phase reaction. Actually, the cell design in operando testing, such as the open window with insulated polymer, the limited electrolyte, the vertical hang of the cell during operando testing, may impact the battery performance and the discharge plateau of the cell in operando test is a little bit lower than the ones in regular test.

ACTION: We updated the in-operando XANES spectra with newly added references in Figure 2d-f in the manuscript. The following description are also added in the manuscript:

“The results of the operando sulfur K-edge XANES with reference samples are presented in Figure 2d-f alongside a schematic outline.”

“The intensity of this peak is found to vary as the electrochemical reaction proceeds, indicating the redox reaction of polysulfides. At the end of discharge process, the linear polysulfide peak becomes almost invisible and the peak of Li_2S appears.”

“It should be noted that the peak of the Li_2S in Figure 2e is not fully reversed back to sulfur during the operando charging process. Actually, the first cycle coulombic efficiency of alucone coated C-S cathode is around 90%, which indicates that some of the Li_2S is not reversed back to sulfur. Therefore, it is normal that at the end of the first charging process in the operando XANES study we can still see the Li_2S peak in the XANES spectrum.”

Comment 3: The authors should provide other supporting data for their claims. For example, providing evidence of no color change in the electrolyte during discharge would be direct evidence for elimination/suppression of polysulfide formation.

Author reply: Thank you for the suggestions. To further demonstrate the elimination of polysulfide, various physicochemical characterizations have been performed and shown in Figure 5.

Figure 5. Physicochemical characterizations to demonstrate the elimination of polysulfides. (a) Electrolyte color test of the Li-S cell with alucone coated C-S electrodes in ether and carbonate electrolytes. (b) TOF-SIMS spectra of cycled alucone coated C-S electrodes in ether and carbonate electrolytes. (c) RBS spectra of the Li metal anodes in different electrolytes after cycling.

Electrolyte color evaluation of Li-S batteries with alucone coated C-S electrodes is performed in Figure 5a. During the electrochemical reaction, the ether-based electrolyte with alucone C-S has changed to light yellow while the carbonate based electrolyte remains constant as colorless liquid, illustrating the elimination of dissolved polysulfides in carbonate based electrolyte. Figure 5b presents time of flight secondary ion mass spectra (TOF-SIMS) of the alucone coated C-S electrodes cycled in ether-based and carbonate-based electrolytes. The unique mass fragments are highlighted and shown in the magnified diagrams. From the negative TOF-SIMS spectra, the alucone coated electrode cycled in ether-based electrolyte presents strong peaks of S^- , S^{2-} , S^{3-} species, indicating the formation of linear polysulfides in the electrochemical process. On the other hand, the electrode cycled in carbonate electrolyte only presents the peak of S^- but the peaks of S^{2-} and S^{3-} are not obvious, further demonstrating the absence of polysulfides during the electrochemical process in carbonate electrolyte. To further confirm that no polysulfides diffused and migrated in the battery, Rutherford backscattering spectrometry (RBS) was performed on the Li metal anodes, as shown in Figure 5c. Obviously, the presences of S peak (blue line) confirms the deposition of polysulfide species on Li anode cycled in ether based electrolyte. However, the spectrum of Li anode cycled in carbonate electrolyte (red line) does not show S peak, indicating no dissolved polysulfide deposited on Li metal anode. All of these supporting characterizations further confirm the elimination of polysulfides in the discussed Li-S batteries with carbonate-based electrolyte.

ACTION: We added Figure 5 as Figure S14 in supporting information. Corresponding description is added in manuscript and supporting information.

In manuscript

“To further demonstrate the elimination of polysulfides, three physical characterizations are carried out and shown in Figure S14, including the observation of electrolyte colors; time of flight secondary ion mass spectrometry (TOF-SIMS) of alucone coated C-S electrodes after cycling; and Rutherford backscattering spectrometry (RBS) of cycled Li metal anodes. All of these supporting

experiments confirmed the elimination of polysulfides in the carbonate based Li-S batteries. Detailed experiments process and results can be seen in supporting information.”

In supporting information

“Electrolyte color evaluation of Li-S batteries with alucone coated C-S electrodes is performed in Figure S14a. During the electrochemical reaction, the ether-based electrolyte with alucone C-S has changed to light yellow while the carbonate based electrolyte remains constant as colorless liquid, illustrating the elimination of dissolved polysulfides in carbonate based electrolyte. Figure S14b presents time of flight secondary ion mass spectra (TOF-SIMS) of the alucone coated C-S electrodes cycled in ether-based and carbonate-based electrolytes. The unique mass fragments are highlighted and shown in the magnified diagrams. From the negative TOF-SIMS spectra, the alucone coated electrode cycled in ether-based electrolyte presents strong peaks of S^- , S^{2-} , S^{3-} species, indicating the formation of linear polysulfides in the electrochemical process. On the other hand, the electrode cycled in carbonate electrolyte only presents the peak of S^- but the peaks of S^{2-} and S^{3-} are not obvious, further demonstrating the absence of polysulfides during the electrochemical process in carbonate electrolyte. To further confirm that no polysulfides diffused and migrated in the battery, Rutherford backscattering spectrometry (RBS) was performed on the Li metal anodes, as shown in Figure S14c. Obviously, the presences of S peak (blue line) confirms the deposition of polysulfide species on Li anode cycled in ether based electrolyte. However, the spectrum of Li anode cycled in carbonate electrolyte (red line) does not show S peak, indicating no dissolved polysulfide deposited on Li metal anode. All of these supporting characterizations further confirm the elimination of polysulfides in the discussed Li-S batteries with carbonate-based electrolyte.”

Comment 4: Presenting XRD of both the alucone coated CS electrode and short chain sulfur cathode could provide evidence of their difference.

Author reply: Thanks for the suggestions. X-ray diffraction (XRD) was carried out for the as-prepared sulfur cathodes. As shown in Figure 6a, sulfur in the both of the electrodes were in an amorphous state, which makes it hard to distinguish the differences between cyclo- S_8 cathode and small sulfur S_x cathode. Hence, a XANES study was also performed to probe the fine structure of as-prepared C-S electrodes in Figure 6b. Compared with the cyclo- S_8 electrode, the small sulfur

electrode presents a higher shoulder at 2473.5 eV, which can be assigned to the transition from S 1s to the C–S σ^* state (*J. Phys. Chem. C* 2016, 120, 10111–10117). The appearance of this shoulder peak indicates a strong interaction between short-chain sulfur and carbon hosts, which has also been reported in previous references related to small sulfur cathodes.

Figure 6. XRD and XANES spectra of the pristine as-prepared sulfur electrodes.

ACTION: We have added the Figure 6b and following description as Figure S4 in supporting information and manuscript, respectively.

“Compared with the cyclo-S₈ electrode, the small sulfur electrode presents a higher shoulder at 2473.5 eV, which can be assigned to the transition from S 1s to the C–S σ^* state. The appearance of this shoulder peak indicates a strong interaction between short-chain sulfur and carbon hosts.”

We also cited one more reference in the manuscript here:

41. Ye Y, *et al.* X-ray Absorption Spectroscopic Characterization of the Synthesis Process: Revealing the Interactions in Cetyltrimethylammonium Bromide-Modified Sulfur–Graphene Oxide Nanocomposites. *J. Phys. Chem. C* **120**, 10111-10117 (2016).

Comment 6: In Figure 5e, two peaks are still observed during lithiation even in carbonate electrolyte. If polysulfide formation is eliminated, why are there still two peaks?

Author reply: Thank you for the question. We assume you refer to the CV profiles shown in Figure 4d in the manuscript. From our perspective, the multi-peaks occurred in the cathodic process is due to the multi porous structure from carbon hosts. Sulfur confined in different porous structures may have different reaction rates with Li-ion and leads to various potential polarization in batteries, especially in a solid-phase Li-S reaction. As a result, multi-cathodic peaks appear in CV profiles.

Figure 7. Effect of porous structure on the performance of Li-S batteries in carbonate electrolyte. (a-c) CV profiles of the Li-S batteries with different carbon hosts. (d) Summary of the three carbon hosts in terms of surface area, average pore size, and corresponding cathodic peaks of sulfur cathodes. (e) CV profiles of Li-S batteries in ether based electrolyte.

To demonstrate our points, Figures 7a-c present the CV profiles of alucone coated sulfur cathodes with three different carbon hosts (BP800, BP1300, and BP2000). Figure 7d summarizes the pore size distribution, surface area, and corresponding cathodic peak potentials of the three carbon hosts. Although the three cathodes performed similar one-plateau discharge-charge profiles, they demonstrate three different cathodic peaks in CV profiles. The reason is due to the different porous structures of the three carbon hosts. As summarized in Figure 7d, the pore structure of BP800 is relative simple which are mainly consisting of large mesopores > 10 nm. The cathodic peak of BP800 at 1.45-1.5 V is lower than that of the two other sulfur cathodes. BP1300, on the other hand, has a more complicated porous structure, which is composed with micropores (< 2 nm), small mesopores (2-10 nm), and few large mesopores (>10 nm). Therefore, in the CV cathodic process, multi-peaks appeared from 1.5-1.75 V, which is the broadest cathodic peak among the three sulfur cathodes. Furthermore, BP2000 has the smallest pore size distribution among the three carbon hosts which is composed with small mesopores (mostly 2-4 nm) and micropores (< 2 nm) and has a cathodic peak potential at 1.6-1.75 V. Therefore, the three CV profiles with different carbon hosts illustrates (1) larger pore size of carbon host induces more severe potential polarization in the solid-phase Li-S electrochemical reaction; (2) smaller porous structure is preferred in the solid-phase Li-S reaction, which facilitates widely scatted sulfur distribution and increases the electronic conductivity of sulfur cathode, as shown in Figure 8; (3) hierarchically porous structure leads to multi-peaks appearing in the cathodic process of CV profiles in the solid-phase Li-S electrochemical reaction.

Figure 8. Schematic figure of the sulfur distribution in different porous structure.

To further illustrate that the multi-peaks do not originate from polysulfides, we also added the Li-S batteries in ether-based electrolyte in Figure 7e. If polysulfides will lead to the multi-peaks appearance, the CV profiles of Li-S batteries in ether based electrolytes should also demonstrate multi-peaks forming a broad peak at 2.1 V. However, as shown in Figure 7e, only one sharp peak at 2.1 V appearance, corresponding to the cathodic process from Li_2S_6 to Li_2S . Therefore, this further demonstrates that the multi-peaks appearance in carbonate electrolyte is not related to the formation of polysulfides.

ACTION: We have added Figure 7 and Figure 8 as Figure S10 and Figure S11 in supporting information and added more description in the manuscript.

In manuscript:

“The differences observed in cycling performance between the three sulfur electrodes are proposed to originate from the nanostructured carbon hosts. Compared to the other carbon hosts, BP2000 has the highest surface area with a confined porous architecture. This unique structure allows for highly dispersed sulfur distribution throughout the host material and improves the electronic conductivity of sulfur in the solid-phase Li-S reaction.”

“Cyclic voltammograms of the three alucone coated C-S cathodes are presented in Figure 4d and Figure S10. All coated cathodes demonstrated reversible Li-S redox electrochemical reaction in carbonate electrolytes, with a single pair of cathodic and anodic peaks at around 1.75 V and 2.3 V, respectively. Compared to the other two carbon hosts as shown in Figure S10, the sulfur cathode with BP2000 exhibited sharp and highly reversible peaks. Furthermore, the cathodic peak potential of the sulfur cathode with BP2000 was also higher than that of others, indicating smaller potential polarization and high electrochemical reaction activity of the Li-S battery.”

Reviewer #3

General comments: This work discusses the mechanisms of Li-S reactions in carbonate based electrolyte by coating S/C with alucone. The author mainly claim two things: 1) carbonate is safer than ether and 2) they demon a "high" S loading electrodes in carbonate electrolytes and project the cell energy will be high. However, these two claims are not fully supported by their work. The above work needs to demonstrate more evidence and discuss why alucone coating can prevent continuous electrolyte attack to polysulfide or promote the direct conversion. They also need to prepare a real pouch cell instead of doing simulation to show their claims are correct. Otherwise, the work cannot meet the qualification of NC.

Author reply: Thank you very much for the questions and suggestions, especially the suggestions of pouch cell, which are very helpful to improve the quality of this paper. In this revision, apart from the pouch cell, we have also carried out more physical characterizations, electrochemical test, and added more discussion and references in the manuscript to answer reviewer's questions.

Comments 1: Although carbonate electrolytes, compared to ether based one, have some advantages such as wider electrochemical window, the cycling of S does not need to go above 3.5 V at which the decomposition of ether begins. More importantly, for lithium metal based batteries (Li-S is a Li metal battery), ether is more benign than carbonate solvents towards lithium metal and SEI generated is much resistance as reported by many literature. DOL/DME indeed has lower vapor pressure and boiling points than carbonate solvents. However, either ether or carbonate, they are both organic solvents and functions as "fuels" once batteries catch fire. From this point of view, there is no big difference between ether or carbonate based electrolytes in terms of safety. Remember, thermal runaway starts from SEI breakdown.

Author reply: That's a very good question! Yes, both ether and carbonate solvents are flammable liquids. However, it should be noted that carbonate based electrolyte have been applied in commercialized Li-ion batteries over decades and many fire retardant additives designed for carbonate electrolytes have been investigated and are applicable (Journal of Power Sources 2005, 146, 116; Journal of Power Sources 2006, 162, 1379). Furthermore, before their application in electric vehicles, commercialized Li-ion batteries with carbonate based electrolytes have been passed the rigid safety evaluation, indicating a wide acceptability of carbonate electrolyte in the Li-ion battery manufacturing industry. Therefore, from a battery fabrication view, it is good to have a

smooth transfer from the traditional Li-ion metal oxide cathodes to high energy sulfur cathodes with the usage of mature electrolyte systems. Therefore, in this work, we have prepared alucone coated C-S electrodes with various carbon hosts and attempted to use various carbonate electrolytes to confirm the universality of the alucone coating method. Furthermore, we reported for the first time, the success operation of Li-S pouch cell in carbonate electrolyte, demonstrating the compatible fabrication of Li-S batteries in a mature Li-ion electrolyte system. We believe this work has the potential to guide future high energy Li-S batteries.

ACTION: We have made some clarifications in the introduction to further explain the importance of the use of carbonate based electrolyte in Li-S batteries.

Carbonate-based electrolyte systems have been used in commercial Li-ion batteries due to their safe and stable properties as well as wide-operational temperature window.^{11, 12} Furthermore, many flame-retardant additives designed for carbonate based electrolyte have been investigated and applied into the battery market.^{13, 14} The stable and smooth transfer from traditional metal-oxide cathodes to sulfur cathode with a mutual carbonate electrolyte system may promote the realization of safe and high-energy Li-S batteries in the future.

We also cited more references here:

13. Doughty DH, Roth EP, Crafts CC, Nagasubramanian G, Henriksen G, Amine K. Effects of additives on thermal stability of Li ion cells. *J Power Sources* **146**, 116-120 (2005).
14. Zhang SS. A review on electrolyte additives for lithium-ion batteries. *J Power Sources* **162**, 1379-1394 (2006).

Comments 2-4: The calculation on the cell energy is not accurate. It is not convincing that with 4 mg/cm² S loading, the authors can reach 350 Wh/kg. The amount of minimum electrolyte needed is determined by the cathode architecture. With carbon paper as the host, a large amount of excessive amount of electrolyte (much more than 3 times) will be required significantly reducing the energy. The cell energy calculation is also based on the assumption that they don't need Cu current collector which is not true. The averaged voltage is much lower than those tested in ether based one which further lowers the energy. **(1)** It is suggested that the authors prepare a pouch cell by using the method they discussed to see if they can reach the number even at the first discharge. In addition, Sion Power already demonstrate the same energy density Li-S pouch cell before in ether based

electrolytes further confirming that it does not matter which electrolyte is used as long as it works. (2) The carbon host used in this work has high surface area so the authors used a different current collector i.e., carbon paper to make the S/C cathodes to avoid delamination. However, the delamination is a common phenomenon for all electrodes consisting of nano particles. In other words, the authors should address the coating issue of using nano materials instead of using carbon paper as the current collector. It will make the comparison with control experiments not at the same level. Carbon paper cannot be used in pouch cell assembly either. (3) It seems that in all the experimental results the authors have no control on the E/S ratio which is key to obtain the best performances of Li-S cells. The inconsistency of current collector used also makes the conclusions a question mark.

Author reply: Thank you for the questions and suggestions for the energy density calculation. In this revision, we have prepared pouch cell as shown in Figure 1. To overcome the issues of carbon paper current collector, we used SBR-CMC as the binder instead of PVDF, which successfully prepared the nanomaterials slurry on commercial Al foil current collector instead of carbon paper. Further, all of as-prepared pouch cells have controlled the electrolyte ratio less than 3:1 to sulfur mass. In our pouch cell, we did not add Cu foil as anode current collector, which the nickel-tap was directly pressed on the Li foil. Detailed pouch cell assembly process and parameters can be found in supporting information.

Figure 1. Pouch cell characterization of alucone coated C-S electrodes in carbonate electrolyte. (a) As-prepared C-S electrodes (before and after MLD coating) and the assembled pouch cell. (b) Discharge-charge profiles of assembled pouch cells under different conditions. (c) Calculated energy density of our work and reported Li-S batteries in carbonate electrolyte. The reported energy density of *Sion Power* Li-S pouch cell in ether based electrolyte is labeled as a reference (*Sion Power* news Available online: <https://sionpower.com/2015/airbus-ds-and-sion-power-enter-into-collaborative-agreement>).

***Notes:** The energy density of *Sion Power* Li-S pouch cell is reported to be $> 350 \text{ Wh kg}^{-1}$ from public information source. Herein, we assumed that this energy density value is based on cell-level and the package mass ratio in the whole pouch cell is around 5 wt% (estimated based on commercial LIBs). So the energy density at electrode level of *Sion Power* can be calculated to be

around 370 Wh kg^{-1} , which is marked in dashed line in the figure to compare. [#]In our actual assembled pouch cell, the thinnest Li foil that we can get is $100 \mu\text{m}$ and the amount of Li is over 840% of sulfur. When using thinner Li foils with the thickness of 18 and $12 \mu\text{m}$ which are corresponding to 100% and 150% of sulfur (the same condition in estimation), the energy density will significantly increase.

Figure 1a presents the prepared electrodes and assembled pouch cell. Discharge-charge profiles are shown in Figure 1b and the detailed parameters of pouch cell assembly are shown in Table 1. **Firstly**, it should be noted that the pouch cells with both single-sided and double-sided electrodes are operated successfully in carbonate electrolyte, which, to our knowledge, is the first reported Li-S pouch cell in carbonate electrolyte. The first discharge capacity can reach over 1100 mAh g^{-1} at 0.05 C . The pouch cell demonstrates good cycling reversibility which the 2nd cycle discharge capacity can maintain over 780 and 430 mAh g^{-1} at 0.002 C and 0.05 C , respectively. The discharge profiles from the second cycle performed one plateau, well confirming the solid-phase Li-S reaction. **Secondly**, it should be noted there are still many challenges in the operation of current pouch cells. A big irreversible capacity can be found from the second cycle. This may be due to **(1)** some of cyclo- S_8 was not covered by alucone coating. We can still see the voltage plateau at 2.3 V from the uncovered cyclo- S_8 in the first cycle. The capacity at this plateau is irreversible and from the second cycle only one discharge plateau appeared; **(2)** limited electrolyte added in the pouch cell. Compared with the coin cell, we controlled the amount of electrolyte adding in pouch cell, which may reduce the Li-ion conductivity of the cell, leading to the decrease of capacity; and **(3)** the electrolyte did not have FEC as used in coin-cells, which may affect the cycling performance.

Table 1. Pouch cell assembly parameters

Electrode size	77 mm * 50 mm
Sulfur host	BP2000
Sulfur content	> 65 wt%
Sulfur loading (After MLD)	1.35-1.45 mg cm^{-2} (Single-sided) 2.65-2.95 mg cm^{-2} (Double-sided)
C-S composite: Binder	9:1
Current collector	Al foil
Electrolyte	LiPF_6 (EC:DEC, v:v=1:1)
Electrolyte: Sulfur	< 3:1

Due to time limitations, we will further continue to improve the pouch cell performance in the future. Meanwhile for now, we believe the success demonstration of the Li-S pouch cell in carbonate electrolyte as a first attempt demonstrates the potential practical application of carbonate based Li-S batteries. The measured and estimated energy density of our results and simulated energy density of reported carbonate based Li-S coin cells are shown in Figure 1c. Detailed calculation parameters are shown in supporting information Table S1, S3, S5, and Figure S15. We also added *Sion Power's* ether-based Li-S pouch cell as a reference in the figures. The energy densities are only calculated at the electrode level without any shell or package. To estimate the energy density of pouch cells, all of reported coin-cell data in previous references are simulated assuming the E/S ratio of 3:1 and adopting the first cycle discharge capacity, which reflects the highest energy density the batteries can reach. In this study, the measured actual pouch cell energy density can reach over 200 Wh kg⁻¹ with a thick Li foil of 100 μm, which the Li amount is over 840% of sulfur. If using a thinner Li foil in the future as the estimation condition (100%-150% of sulfur), the energy density can be improved to over 280 Wh kg⁻¹. Compared with the simulated pouch cell energy density of all previous reported carbonate Li-S cells, both our pouch cell and coin cell results exhibit a competitive energy density. We believe that our present study demonstrates the potential of future carbonate-electrolyte based Li-S batteries to compete the high energy density as the Li-S cell in ether based electrolyte.

ACTION: We have added Figure 1 and Table 1 in the manuscript as Figure 5 and supporting information as Table S4, respectively. We also added the following description in the manuscript:

“To further probe the potential practical application of the developed carbonate electrolyte Li-S batteries, pouch cell characterizations were performed. The detailed parameters of pouch cell assembly are shown in Table S4. Figure 5a presents the prepared electrodes and assembled pouch cell. Discharge-charge profiles are shown in Figure 5b. Firstly, it should be noted that the pouch cells with both single-sided and double-sided electrodes are operated successfully in carbonate electrolyte, which, to our knowledge, is the first reported Li-S pouch cell in carbonate electrolyte. The first discharge capacity can reach over 1100 mAh g⁻¹ at 0.05 C. The pouch cell demonstrates good cycling reversibility which the 2nd cycle discharge capacity can maintain over 780 and 430 mAh g⁻¹ at 0.002 C and 0.05 C, respectively. The discharge profiles from the second cycle performed one plateau, well confirming the solid-phase Li-S reaction. Secondly, it should be noted there are still many challenges in the operation of current pouch cells. A big irreversible capacity can be found from the second cycle. This may be due to (1) some of cyclo-S₈ was not covered by

alucone coating. We can still see the voltage plateau at 2.3 V from the uncovered cyclo-S₈ in the first cycle. The capacity at this plateau is irreversible and from the second cycle only one discharge plateau appeared; (2) limited electrolyte added in the pouch cell. Compared with the coin cell, we controlled the amount of electrolyte adding in pouch cell, which may reduce the Li-ion conductivity of the cell, leading to the decrease of capacity; (3) the electrolyte did not have FEC as used in coin-cells, which may affect the cycling performance.

The success of the Li-S pouch cell in carbonate electrolyte as a first attempt demonstrates the potential practical application of carbonate based Li-S batteries. The measured and estimated energy density of our results and simulated energy density of reported carbonate based Li-S coin cells are shown in Figure 5c. Detailed calculation parameters are shown in supporting information Table S1, S3, S5, Figure S15. We also added *Sion Power's* ether-based Li-S pouch cell as a reference in the figures. The energy densities are only calculated at the electrode level without any shell or package. To estimate the energy density of pouch cells, all of reported coin-cell data in previous references are simulated assuming the E/S ratio of 3:1 and adopting the first cycle discharge capacity, which reflects the highest energy density the batteries can reach. In this study, the measured actual pouch cell energy density can reach over 200 Wh kg⁻¹ with a thick Li foil of 100 μm, which the Li amount is over 840% of sulfur. If using a thinner Li foil in the future as the estimation condition (100%-150% of sulfur), the energy density can be improved to over 280 Wh kg⁻¹. Compared with the simulated pouch cell energy density of all previous reported carbonate Li-S cells, both our pouch cell and coin cell results exhibit a competitive energy density. We believe that our present study demonstrates the potential of future carbonate-electrolyte based Li-S batteries to compete the high energy density as the Li-S cell in ether based electrolyte.”

Comments 5: The only evidence used to prove that after alucone coating, the S₈ directly converts to L2S is XANES. However, the lacking of standard reference materials during the transition leads to many uncertainties. If what the authors claimed is true, then why in all solid state Li-S, still two clear plateaus are seen similar as in liquid cells?

Author reply: Thank you for the questions on the XANES results and solid-phase Li-S transformation. To remove the reviewer's doubt of the XANES results, we have added the standard samples, sulfur powder, Li₂S powder, as well as prepared Li₂S₆ and Li₂S₄ solutions as references in

Figure 2. Furthermore, we have also updated Figure 2d in the manuscript with **newly acquired operando XANES data for better spectra quality**. This time we reduced the amount of electrolyte and fortunately got better quality of spectra to clearly observe the evolution of the sulfur cathode in ether-based electrolyte, as shown in Figure 2d. The spectra has demonstrated a feature at around 2470 eV belonging to chain polysulfides S_x^{2-} . The intensity of this feature was changing along with the discharge-charge process, indicating the redox reaction of chain polysulfides. The three operando XANES comparison with standard references illustrates the formation of polysulfides of the Li-S battery in ether based electrolyte (Figure 2d), while the other two (Figure 2e-f) in carbonate based electrolyte do not demonstrate peaks of linear polysulfides during the whole discharge-charge process, indicating a direct transformation of sulfur to Li_2S .

Figure 2 (Figure 5d-f in manuscript). In-operando XANES study of (d) Alucone coated C-S electrode in ether-based electrolyte, (e) alucone coated C-S electrode in carbonate-based electrolyte, and (f) as-prepared small sulfur molecule cathode in carbonate-based electrolyte

The second part of the question from the reviewer is the possibility of solid-phase Li-S transformation. Actually, the viewpoint that all-solid-state Li-S batteries (SS Li-S) are still observed two clear plateaus which is similar as in liquid cells is not completely correct. Up to now, there are two types of all-solid-state Li-S batteries based on the different types of solid-state electrolytes, polymer and inorganic (sulfide) electrolytes. Many (but not all) polymer based SS Li-S batteries

show the two plateaus in the reaction. Taking the typical solid-state polymer, polyethylene glycol (PEO) for example, most reported PEO SS Li-S batteries still need to operate the cell at 60-70 °C to make the PEO polymer electrolyte in a semi-melting state to have enough ionic conductivity (*J. Phys. Chem. Lett.* 2017, 8, 3473; *J. Mater. Chem. A*, 2017, 5, 12934; *J. Phys. Chem. Lett.* 2017, 8, 1956, *J. Power Sources* 2016, 319, 247). The melted polymer electrolyte has a liquid property and PEO structure actually is very similar to dimethoxyethane solvent (DME). Therefore there is no doubt the polysulfides can be dissolved into melting PEO polymer and the Li-S cell still undergoes liquid-solid dual-phase reaction like in ether-based electrolyte.

Figure 3. Comparison of two types of all solid-state Li-S batteries (Electrochem. Energy Rev. DOI: 10.1007/s41918-018-0010-3)

On the other hand, the other major type of SS Li-S batteries is with the use of sulfide based electrolyte, such as, $P_2S_5-Li_2S$, Li_6PS_5Cl , $Li_{10}GeP_2S_{12}$ (*J. Electrochem. Soc.* 2015 162, A646; *Nano Lett.*, 2016, 16, 4521; *Adv. Energy Mater.* 2017, 7, 1602923). With a high ionic conductivity, the reported sulfide based SS Li-S batteries can be operated at room temperature. All the sulfur cathodes in these references using inorganic solid-state electrolytes demonstrated one-plateau discharge-charge process during the Li-S redox reaction, corresponding to the solid-phase conversion of S to Li_2S , which is consistent with our results. The whole electrochemical reaction in sulfide based Li-S

batteries is in solid-state without the dissolution and immigration of polysulfides. Therefore, the solid-phase Li-S transformation exists and highly depends on the configurations of Li-S system.

ACTION: We updated the in-operando XANES spectra with newly added references in Figure 2d-f in the manuscript. The following description are also added in the manuscript:

“The results of the operando sulfur K-edge XANES with reference samples are presented in Figure 2d-f alongside a schematic outline.”

“The intensity of this peak is found to vary as the electrochemical reaction proceeds, indicating the redox reaction of the polysulfides. At the end of the discharge process, the linear polysulfide peak becomes nearly invisible and the peak of Li_2S appears.”

Comments 6: The addition of FEC enhances SEI quality on Li. The authors did not discuss what are the interlaces in figure 6S. in carbonate electrolytes, did authors still see Sulfur cross over to the anode side? Or Sulfur redistribution? The main advantage of the proposed mechanism is the direct conversion and avoid the generation of soluble species. Did shuttle still happen though? All these critical questions were not mentioned.

Author reply: Thank you very much for the important suggestions. To further demonstrate the elimination of polysulfide, various physicochemical characterizations have been performed and shown in Figure 4.

Figure 4. Physicochemical characterizations to demonstrate the elimination of polysulfides. (a) Electrolyte color test of the Li-S cell with alucone coated C-S electrodes in ether and carbonate electrolytes. (b) TOF-SIMS spectra of cycled alucone coated C-S electrodes in ether and carbonate electrolytes. (c) RBS spectra of the Li metal anodes in different electrolytes after cycling.

Electrolyte color evaluation of Li-S batteries with alucone coated C-S electrodes is performed in Figure 4a. During the electrochemical reaction, the ether-based electrolyte with alucone C-S has changed to light yellow while the carbonate based electrolyte remains constant as colorless liquid, illustrating the elimination of dissolved polysulfides in carbonate based electrolyte. Figure 4b presents time of flight secondary ion mass spectra (TOF-SIMS) of the alucone coated C-S electrodes cycled in ether-based and carbonate-based electrolytes. The unique mass fragments are highlighted and shown in the magnified diagrams. From the negative TOF-SIMS spectra, the alucone coated electrode cycled in ether-based electrolyte presents strong peaks of S^- , S^{2-} , S^{3-} species, indicating the formation of linear polysulfides in the electrochemical process. On the other hand, the electrode cycled in carbonate electrolyte only presents the peak of S^- but the peaks of S^{2-} and S^{3-} are not obvious, further demonstrating the absence of polysulfides during the electrochemical process in carbonate electrolyte. To further confirm that no polysulfides diffused and migrated in the battery, Rutherford backscattering spectrometry (RBS) was performed on the Li metal anodes, as shown in Figure 4c. Obviously, the presences of S peak (blue line) confirms the deposition of polysulfide species on Li anode cycled in ether based electrolyte. However, the spectrum of Li anode cycled in carbonate electrolyte (red line) does not show S peak, indicating no dissolved polysulfide deposited

on Li metal anode. All of these supporting characterizations further confirm the elimination of polysulfides in the discussed Li-S batteries with carbonate-based electrolyte.

ACTION: We added Figure 4 as Figure S14 in supporting information. Corresponding description is added in manuscript and supporting information.

In manuscript

“To further demonstrate the elimination of polysulfides, three physical characterizations are carried out and shown in Figure S14, including the observation of electrolyte colors; time of flight secondary ion mass spectrometry (TOF-SIMS) of alucone coated C-S electrodes after cycling; and Rutherford backscattering spectrometry (RBS) of cycled Li metal anodes. All of these supporting experiments confirmed the elimination of polysulfides in the carbonate based Li-S batteries. Detailed experiments process and results can be seen in supporting information.”

In supporting information

“Electrolyte color evaluation of Li-S batteries with alucone coated C-S electrodes is performed in Figure S14a. During the electrochemical reaction, the ether-based electrolyte with alucone C-S has changed to light yellow while the carbonate based electrolyte remains constant as colorless liquid, illustrating the elimination of dissolved polysulfides in carbonate based electrolyte. Figure S14b presents time of flight secondary ion mass spectra (TOF-SIMS) of the alucone coated C-S electrodes cycled in ether-based and carbonate-based electrolytes. The unique mass fragments are highlighted and shown in the magnified diagrams. From the negative TOF-SIMS spectra, the alucone coated electrode cycled in ether-based electrolyte presents strong peaks of S^- , S^{2-} , S^{3-} species, indicating the formation of linear polysulfides in the electrochemical process. On the other hand, the electrode cycled in carbonate electrolyte only presents the peak of S^- but the peaks of S^{2-} and S^{3-} are not obvious, further demonstrating the absence of polysulfides during the electrochemical process in carbonate electrolyte. To further confirm that no polysulfides diffused and migrated in the battery, Rutherford backscattering spectrometry (RBS) was performed on the Li metal anodes, as shown in Figure S14c. Obviously, the presences of S peak (blue line) confirms the deposition of polysulfide species on Li anode cycled in ether based electrolyte. However, the spectrum of Li anode cycled in carbonate electrolyte (red line) does not show S peak, indicating no dissolved polysulfide deposited

on Li metal anode. All of these supporting characterizations further confirm the elimination of polysulfides in the discussed Li-S batteries with carbonate-based electrolyte.”

Thanks very much again to the reviewers for reviewing our manuscript, and providing constructive questions, comments and suggestions! They are all very important to this paper. We really appreciate it!

REVIEWERS' COMMENTS:

Reviewer #1 (Remarks to the Author):

This revised manuscript reports that Li-S in a carbonate-based electrolyte can be operated by solid-solid transformation without forming lithium polysulfides by using in-operando XAS. The revised manuscript provides more rigorous supporting data and new pouch cell data. However, I hesitate to recommend the publication of this manuscript, at least in its present form. There are some aspects of the work that need to be addressed.

1. I think that the revised manuscript should further emphasize the new insights from this study compared to the previous report (Li, X., Lushington, A., Sun, Q., Xiao, W., Liu, J., Wang, B., et al. (2016). Safe and Durable High-Temperature Lithium-Sulfur Batteries via Molecular Layer Deposited Coating. *Nano Letters*, 16(6), 3545–3549. <http://doi.org/10.1021/acs.nanolett.6b00577>) to re-arrange the data. Some of data in this study can be similar with the previous paper.
2. This study shows that the change in the phase transformation pathway does not matter with the molecular format of the sulfur. Please speculate the reason why the alucone coated Li-S shows different phase transformation pathway. Do you think that the reaction of LiPF₆ can be matter with this? Have you changed the salt from LiPF₆ to others such as LiClO₄ or LiBF₄?

3. Please polish English

Reviewer #3 (Remarks to the Author):

Revision is OK.

Response to Reviewers' Comments (Manuscript ID NCOMMS-17-34346A)

We are very grateful for the reviewers' constructive comments and insightful suggestions for improving the quality of this manuscript. The major changes in this revision are:

1. We have revised the introduction and conclusion sections to highlight the new insights and innovations of this paper.
2. More experiments were carried out to further reveal the effect of Li salts from electrolytes in carbonate Li-S batteries.
3. The language was further polished in the manuscript
4. More recent related references were cited and discussed.

Reviewer #1

General comments: This revised manuscript reports that Li-S in a carbonate-based electrolyte can be operated by solid-solid transformation without forming lithium polysulfides by using in-operando XAS. The revised manuscript provides more rigorous supporting data and new pouch cell data. However, I hesitate to recommend the publication of this manuscript, at least in its present form. There are some aspects of the work that need to be addressed.

Author reply: Thank you very much for your constructive suggestions. In this revision, we have revised the introduction and conclusion sections to highlight the new insights and innovations of this paper. Furthermore, we also carried out more experiments to further reveal the effect of Li salts from electrolytes in carbonate Li-S batteries based on the reviewer's suggestion. The language was further polished in the manuscript, while more recent related references were cited and discussed.

Comment 1: I think that the revised manuscript should further emphasize the new insights from this study compared to the previous report (Li, X., Lushington, A., Sun, Q., Xiao, W., Liu, J., Wang, B., et al. (2016). Safe and Durable High-Temperature Lithium–Sulfur Batteries via Molecular Layer Deposited Coating. *Nano Letters*, 16 (6), 3545–3549. <http://doi.org/10.1021/acs.nanolett.6b00577>) to re-arrange the data. Some of data in this study can be similar with the previous paper.

Author reply: Thank you very much for your constructive suggestion! We believe that our present paper has two main new insights comparing with our previous report:

1. For the first time, in-operando XANES characterization reveals the different electrochemical reaction mechanisms of Li-S batteries in carbonate and ether based electrolyte. The research indicates that the key of Li-S batteries in carbonate electrolyte is not the molecular formation of the sulfur but the solid-phase Li-S reaction, which sulfur transfers to Li_2S directly without the formation of polysulfides.

2. Based on the revealed mechanism, high loading sulfur cathodes and Li-S pouch cells in carbonate electrolyte were developed with promising performance, which demonstrates the potential of carbonate Li-S batteries for practical application. To our best knowledge, it is the first public reported carbonate-electrolyte based Li-S pouch cell. The development of high loading sulfur cathodes and Li-S pouch cells are a revolutionary breakthrough to the traditional low content sulfur

based composites in carbonate-based electrolyte, which paves a new direction for safe and high energy Li-S batteries in the future.

Accordingly, we believe this research will provide a new direction to realize high-energy Li-S batteries and attract researchers' interests in both electrochemical reaction mechanism of Li-S batteries and surface modification of sulfur cathodes in the future. As suggested by the reviewer, we have rewritten the *Introduction* and *Conclusion* sections of the revised manuscript to further emphasize these innovations. Truly appreciate for your suggestions again!

ACTION: We emphasized these two new insights in both introduction and conclusion sections in the manuscript and cited more references.

Introduction section:

“In this study, synchrotron based in-operando X-ray absorption near-edge spectroscopy (XANES) is conducted to elucidate detailed mechanisms of Li-S batteries operated in both ether and carbonate-based electrolytes. Compared to conventional ether-based electrolytes, in-operando XANES reveals a drastically different electrochemical reaction pathway for Li-S cells in carbonate-based electrolyte. Interestingly, formation of linear polysulfides was absent in the spectra, suggesting a direct solid-phase transition of sulfur (both cyclo-S₈ and short-chain sulfur) to Li₂S. This fundamental mechanism study indicates that the success of Li-S batteries in carbonate-based electrolyte is not determined by the allotrope of sulfur but rather by the electrochemical reaction pathway undertaken. Based on the reaction mechanisms elucidated in this study, conventional carbon-sulfur (C-S) electrodes with cyclo-S₈ molecule are developed and the electrodes display excellent electrochemical performance in a wide temperature range from -22 to 55 °C. Furthermore, sulfur cathodes with a high sulfur content (67 wt% in sulfur composites) and high areal loading (4.0 mg cm⁻²) exhibit stable capacity over 300 cycles. In particular, we conducted the pouch cell test of Li-S battery in carbonate-based electrolyte and measured the energy density. The development of high loading sulfur cathodes and Li-S pouch cells are a revolutionary breakthrough to the traditional low content sulfur cathodes in carbonate-based electrolyte, paving a new future for the development of safe and high energy Li-S batteries.”

Conclusion section:

“In summary, this research reveals the underlying mechanism of Li-S batteries in carbonate electrolyte and promotes their practical application. *Firstly, a detailed mechanism study is presented to unravel the key factors that govern the reversibility of Li-S batteries in carbonate electrolyte systems.* In-operando XANES suggests a solid-phase Li-S redox reaction taking place in carbonate electrolyte that involves direct transformation between sulfur and Li₂S without the formation of linear polysulfides. This novel mechanism indicates that the molecular format of sulfur is not a limiting factor for achieving highly reversible Li-S batteries in carbonate electrolyte. The significance of using cyclo-S₈ in carbonate electrolytes is to open opportunities for the practical application of Li-S batteries. *Secondly, based on the revealed mechanism, we demonstrate the universality of alucone coating for a variety of sulfur cathodes in carbonate electrolyte.* By optimizing the electrolyte and carbon hosts, sulfur cathodes presents promising electrochemical performance in the developed Li-S batteries and are found to be highly reversible across a wide temperature window of -20 to 55 °C. Furthermore, the sulfur cathodes represent a high sulfur content (67 wt% of composites) and loading (4.0 mg cm⁻²). In particular, the research demonstrates that the Li-S pouch cells can reversible operate in carbonate electrolyte systems, indicating strong potential for practical application. This research sheds light on the use of in-operando XANES to reveal intricate reaction mechanisms of Li-S batteries and to streamline the development of high performance C-S cathodes. We hope the revelation of solid-phase reaction mechanism will trigger increased research interests in high energy Li-S batteries and promote novel electrode architectures for energy storage systems.”

We also added several **more recent references** with more discussion to the revised manuscript:

“It should be noted that the reactions between sulfur species and carbonate or ether-based electrolytes are very complicated and not totally understood. This area of research should be further investigated with theoretical calculations in the future.^{45, 46, 47}”

45. Lee CW, *et al.* Directing the Lithium-Sulfur Reaction Pathway via Sparingly Solvating Electrolytes for High Energy Density Batteries. *ACS Cent. Sci.* **3**, 605-613 (2017).

46. Lang SY, Xiao RJ, Gu L, Guo YG, Wen R, Wan LJ. Interfacial Mechanism in Lithium-Sulfur Batteries: How Salts Mediate the Structure Evolution and Dynamics. *J. Am. Chem. Soc.* **140**, 8147-8155 (2018).
47. Eshetu GG, *et al.* Ultrahigh Performance All Solid-State Lithium Sulfur Batteries: Salt Anion's Chemistry-Induced Anomalous Synergistic Effect. *J. Am. Chem. Soc.* **140**, 9921-9933 (2018).

Comment 2: This study shows that the change in the phase transformation pathway does not matter with the molecular format of the sulfur. Please speculate the reason why the alucone coated Li-S shows different phase transformation pathway. Do you think that the reaction of LiPF₆ can be matter with this? Have you changed the salt from LiPF₆ to others such as LiClO₄ or LiBF₄?

Author reply: Thank you very much for these insightful questions! Actually, we believe that the relations between sulfur/sulfide species and carbonate electrolyte are very complicated, which certainly needs in-depth reaction mechanism study in the future. In order to investigate whether Li salt is the decisive factor of the Li-S reaction reversibility in carbonate electrolyte as the reviewer pointed out, we conducted the CV test of Li-S batteries with two different Li salts (LiTFSI and LiClO₄, 1M, EC: DEC=1:1) with alucone coated C-S electrodes.

As shown in Figure 1, the two Li-S batteries with different electrolytes also demonstrate reversible one-pair redox reactions, indicating the occurring of solid-state S-Li₂S conversion as observed in carbonate electrolyte with LiPF₆ salt. We also noted that some recent research indicates that the electrolyte components, both solvents and Li salts, will impact the electrochemical behavior of Li-S batteries. (*ACS Cent. Sci.* **3**, 605-613 (2017). *J. Am. Chem. Soc.* **140**, 8147-8155 (2018). *J. Am. Chem. Soc.* **140**, 9921-9933 (2018).) From the CV results in Figure 1, the two cells with different Li salts demonstrate very similar cathodic-anodic peaks in CV profiles. It may be concluded that Li salts inside electrolyte may impact the performance of Li-S batteries in carbonate electrolyte rather than directly determine the reaction route of the Li-S batteries in this study.

Figure 1. CV profiles of Li-S batteries in carbonate electrolyte with different Li salts.

Combining all of experiment we conducted, we believe that MLD alucone coating is the key to alter the reaction mechanism and realize the reversibility of Li-S battery with carbonate electrolyte. The mechanism should be contributed by its roles of physically blocking the contact of electrolyte and sulfur, avoiding the formation of polysulfide which is shown unstable in carbonate electrolyte and causes the failure of the cell, and thus forcing the battery to a solid-phase Li-S redox reaction. Since the solid-phase Li-S reaction is perfectly circumvent the formation of lithium polysulfides (in-operando XANES results, Figure 2), which is unstable in carbonate electrolyte (Supplementary Figure 7), the alucone coated C-S electrodes can be well operated in carbonate electrolyte. Nonetheless, it is also worth-noting that the detailed nature of the inner reaction mechanisms, such as the transfer pathway of S_8 to Li_2S as well as the decomposition reaction of polysulfide against carbonate electrolyte, is still not totally understood, which needs in-depth investigation with theoretical calculation in the future. After all, we believe that our present study can be inspiring to these following research works toward future carbonate electrolyte-based Li-S batteries.

Comment 3: Please polish English

Author reply: Thank you very much for the suggestions. We have polished the language of the manuscript. It's really helpful to improve our paper!

Reviewer #3

General comments: Revision is OK.

Author reply: Thank you very much for your kind recommendation!

Thanks very much again to the reviewers for reviewing our manuscript, and providing constructive comments and suggestions! We really appreciate it.